

# Supersaturation, buoyancy, and moist convective dynamics

Wojciech W. Grabowski and Hugh Morrison

Mesoscale and Microscale Meteorology Laboratory, National Center for Atmospheric Research,
 Boulder, CO 80307, USA

*Correspondence to*:  W. W. Grabowski (grabow@ucar.edu)

**Abstract.** Motivated by recent discussions concerning differences of convective dynamics in polluted and pristine environments, the so-called convective invigoration in particular, this paper provides an analysis of factors affecting convective updraft buoyancy, such as the in-cloud supersaturation, condensate and precipitation loading, and entrainment. We use the deep convective period from simulations of daytime convection development over land discussed in our previous publications. An entraining parcel framework in used in the theoretical analysis. We show that for the specific case considered here finite

(positive) supersaturation noticeably reduces pseudo-adiabatic parcel buoyancy and cumulative CAPE in the lower troposphere. This comes from keeping a small fraction of the water vapor in a supersaturated state and thus reducing the latent heating. Such a lower-tropospheric impact is comparable to the effects of the condensate loading and entrainment in the idealized parcel framework. For the entire tropospheric depth, loading and entrainment have a much more significant impact on the total CAPE. For instance, an increase in the fractional entrainment rate from 0.05 km$^{-1}$ to 0.3 km$^{-1}$ reduces the theoretical

level of neutral buoyancy from the upper to the middle troposphere and CAPE by a factor of 4. For the cloud model results, we compare ensemble simulations applying either a bulk microphysics scheme with saturation adjustment or a more comprehensive double-moment scheme with supersaturation prediction. The diagnosed bulk fractional entrainment rate, independent of the microphysics scheme applied in the simulations, is either 0.13 or 0.20 km$^{-1}$ depending on whether we consider profiles of the upper end of the percentile range or of the mean in-cloud equivalent potential temperature. We compare

deep convective updrafts, buoyancies, and supersaturations from all ensembles. In agreement with the parcel analysis, the saturation adjustment scheme provides noticeably stronger updrafts in the lower troposphere. For the simulations predicting supersaturation, there are small differences between pristine and polluted conditions below the freezing level that are difficult to explain by standard analysis of the in-cloud buoyancy components. By applying the piggybacking technique, we show that the lower-tropospheric buoyancy differences between pristine and polluted simulations come from a combination of

temperature (i.e., latent heating) and condensate loading differences that work together to make polluted buoyancies and updraft velocities slightly larger when compared to their pristine analogues. Overall, the effects are rather small and contradict previous claims of a significant invigoration of deep convection in polluted environments.



# 1 Introduction

In the presence of gravity, density differences within a fluid give rise to the Archimedean buoyancy force that drives fluid vertical motions. The magnitude of the buoyancy force per unit mass – the buoyancy for short – is expressed as $g(\rho - \rho_o)/\rho_o$, where $g$ is the acceleration of gravity, and $\rho$ and $\rho_o$ are the densities of the volume of fluid under consideration and the reference (environmental) fluid density, respectively. The buoyancy of cloudy air depends on the air temperature and pressure, water

vapor content, and mass of all cloud and precipitation particles within the volume. It is typically expressed through the so-called density temperature or density potential temperature (see section 4.3 in Emanuel 1994). For the case of the anelastic model applied in numerical simulations discussed in this paper, the buoyancy $B$ is defined as (see Eq. 5 in Grabowski and Smolarkiewicz 2002):

$$B = g\left[(\Theta - \Theta^e)/\Theta_0 + \varepsilon\,(q_v - q_v^e) - q\right] \qquad (1)$$

where $\Theta$ and $q_v$ are the potential temperature and water vapor mixing ratio, $\Theta^e$ is the hydrostatically-balanced environmental potential temperature profile, $q_v^e$ is the corresponding environmental water vapor mixing ratio profile, $\Theta_0$ is the base state anelastic potential temperature profile, $q$ is the sum all condensate and precipitation liquid and ice mixing ratios, and $\varepsilon = R_v$

$/R_d$ - 1 $\approx$ 0.6 ($R_v$ and $R_d$ are the gas constants for water vapor and dry air, respectively). The base-state and environmental profiles are typically derived from the initial sounding used in the simulations. The three terms in the square bracket on the r.h.s. of (1) are referred to as the temperature, virtual, and mass-loading terms.

There are two key processes that affect cloudy air buoyancy. The first is the phase change of the water substance that modifies

the air temperature and can alter the virtual and mass-loading terms. For instance, condensation of $\Delta = 1$ g kg$^{-1}$ of water vapor mixing ratio changes the air temperature by $\sim L_v/c_p\,\Delta$ or about 2.5 K and contributes $\sim 0.01g$ to the cloud buoyancy ($L_v = 2.5 \times 10^6$ J kg$^{-1}$ is the latent heat of condensation, $c_p = 1005$ J kg$^{-1}$ K$^{-1}$ is the specific heat of air at constant pressure). Condensation reduces the virtual term by about $0.0006g$ and increases the loading term by $0.001g$. Hence, the buoyancy increase due to temperature change is about an order of magnitude larger than the other terms. For the liquid to ice phase change, freezing of

$\Delta = 1$ g kg$^{-1}$ of liquid water mixing ratio changes the air temperature by $L_f/c_p\,\Delta$ or about 0.3 K and contributes $\sim 0.001g$ to the buoyancy with no effect on virtual and mass-loading terms ($L_f = 3 \times 10^5$ J kg$^{-1}$ is the latent heat of freezing). The second key process affecting cloudy air buoyancy concerns precipitation. Precipitation allows the condensate $q$ in (1) to leave the volume (the condensate off-loading) and thus increases the volume buoyancy. For instance, converting of $\Delta = 1$ g kg$^{-1}$ of cloud condensate (cloud water or ice) into precipitation (rain or snow) and off-loading it increases the volume buoyancy by $\sim 0.001g$,

that is, similar to the impact of freezing it. The similar contributions to the buoyancy of the latent heat of freezing and of the mass loading is the crux of the argument against the hypothesized convection invigoration in polluted environments. The



invigoration argument alleges that carrying the liquid water across the 0ºC level (rather than converting it to rain and off-loading below the freezing level) and freezing it aloft leads to stronger upper-tropospheric updrafts in the polluted environments; see the discussion in section 2a of Grabowski and Morrison (2020, GM20 hereafter, and references therein), and the exchange between Fan and Khain (2021) and Grabowski and Morrison (2021). Precipitation can also fall from above into the volume under consideration and reduce its buoyancy.

There is also a more subtle effect related to the presence of the supersaturation in natural clouds.[1] Positive supersaturation reduces the latent heating when compared to the situation with no supersaturation because some water vapor is left in the supersaturated state. This leads to a decrease of the cloud buoyancy because latent heating dominates the impact on the buoyancy as illustrated above. The impact of the latent heating difference between pristine and polluted clouds is referred to in Fan et al. (2018) as "warm-phase invigoration". Grabowski and Jarecka (2015) derived the density potential temperature difference between situations with finite and zero supersaturations. The difference is approximately a linear function of the supersaturation and depends on the temperature and pressure. In the lower troposphere, supersaturation of 0.01 (i.e., 1%) reduces the density temperature by about 0.1 K (see Fig. 1 in Grabowski and Jarecka 2015), that is, by ~$0.0003g$. However, if the supersaturation ratio is much larger, say 0.1 (i.e., 10%), the impact becomes comparable to the loading of a few g kg$^{-1}$ of the cloud or precipitation mixing ratio. The impact gets smaller in the middle and upper troposphere because the same supersaturation as in the lower troposphere translates into a smaller absolute difference in the condensation rate between the finite and zero supersaturation with lower temperature (cf. Fig. 2 in Grabowski and Morrison 2017).

Finally, buoyancy is strongly affected by entrainment of environmental air and cloud dilution. Entrainment has been argued to affect mean cloud properties from early days of cloud dynamics based on theory and observation (e.g., Stommel 1947, Warner 1955). For shallow convection, entrainment typically leads to buoyancy reversal, that is, replacing the positive buoyancy from condensation inside an undiluted updraft with a negative buoyancy resulting from the cloud water evaporation as a result of entrainment (see Grabowski 1993 and references therein). For deep unorganized convection, high-resolution (LES-type) simulations have shown considerable dilution of updraft core properties (e.g., Kuang and Bretherton 2006; del Genio and Wu 2010; Peters et al. 2020) despite having generally smaller fractional entrainment rates than shallow cumulus (de Rooy et al. 2013). Fractional entrainment rates (as well as environmental relative humidity) control the dilution of core buoyancy and hence strongly influence updraft velocities (de Rooy and Siebesma 2010; Morrison 2017; Peters et al. 2020). In this way, entrainment rate also has a dominant influence on the shallow-to-deep transition and ultimately the height attained by moist updrafts (e.g., Khairoutdinov and Randall 2006; Kuang and Bretherton 2006; Morrison et al. 2021). Although the critical

---

[1] Throughout this paper the supersaturation is defined with respect to liquid water saturation, that is $S=q_v/q_{vs}$ -1, where $q_v$ is the water vapor mixing ratio and $q_{vs}$ is its saturated value with respect to the plain water surface.



impact of entrainment on cumulus updraft properties is well accepted, to our knowledge there has been no work analyzing how entrainment impacts the buoyancy changes driven by finite supersaturations.

A traditional cloud physics view is that supersaturations in natural clouds are small, say, a fraction of 1%, except near the cloud base when activation of cloud condensation nuclei (CCN) takes place (e.g., Pruppacher and Klett 1997). Supersaturations cannot be measured directly but can be estimated by assuming a balance between supersaturation source due to local updraft and supersaturation sink due to growth of cloud droplets. The supersaturation estimated in this way is referred to as the quasi-equilibrium supersaturation (Squires 1952). The quasi-equilibrium supersaturation provides an accurate estimate of the in-

cloud supersaturation as long as the phase relaxation time of the droplet population (that depends on the droplet mean radius and concentration) is short compared to the time scales characterizing changes of the droplet population and the cloud updraft (see, for instance, the appendix in Grabowski and Morrison 2021). Estimation of the quasi-equilibrium supersaturation in relatively weak convective clouds featuring gentle updrafts and insignificant precipitation agrees with the traditional view (e.g., Politovich and Cooper 1988). Measurements of the local updraft strength and droplet spectral characteristics in deep

convective clouds with strong updrafts and significant precipitation are difficult using an instrumented aircraft because of flight safety. Prabha et al. (2011, Fig. 9 therein) document observations from pre-monsoon and monsoon deep convective clouds over the Indian subcontinent with updrafts up to about 10 m s$^{-1}$ and corresponding quasi-equilibrium supersaturations up to several percent. The condensation rate inside a rising adiabatic parcel featuring the quasi-equilibrium supersaturation depends on the vertical velocity alone and is independent of the droplet concentration and radius (see section 2a in GM20).

Thus, differences in the condensation rate between polluted and pristine clouds can only occur by an updraft change (resulting from a difference in the cloud buoyancy), a change in entrainment and mixing, or by supersaturation being different from its quasi-equilibrium value.

From the early days of warm (ice-free) cloud modelling, a typical approach has been to assume that clouds always maintain

saturation (e.g., Morton 1957; Ogura 1963; Orville 1965; Soong and Ogura 1973). For dynamic cloud models, such a computationally efficient approach for calculating cloud condensation and evaporation is referred to as saturation adjustment and has been applied in both compressible (e.g., Klemp and Wilhelmson 1978) and anelastic (e.g., Clark 1979, Lipps and Hemler 1982) models. With the advent of more complicated warm-rain microphysics schemes, such as the double-moment (Morrison and Grabowski 2007 and references therein) and spectral (bin) microphysics (e.g., Kogan 1991; Feingold et al.

1996) that allow estimation of the droplet concentration and mean radius, schemes began predicting in-cloud supersaturation. Khain and Lynn (2009), Lebo and Seinfeld (2011) and Lebo et al. (2012) compared supercell splitting simulations applying bulk saturation-adjustment schemes and saturation-prediction bin microphysics. Khain and Lynn (2009) applied the bin microphysics together with the Thompson scheme (Thompson et al. 2004) and showed almost a doubling the maximum vertical velocity for the bulk scheme (35 for bin versus 65 m s$^{-1}$ for bulk, see Fig. 2 there) and doubling of the surface rain accumulation

(see Fig. 5 there). Such differences are unlikely because of the simulated supersaturation alone and have to come from different



reasons, for instance, different organization of deep convective cells simulated by the two schemes as suggested by maps of the surface rain accumulations. Lebo and Seinfeld (2011) applied a similar simulation framework to compare supercell simulations using their bin microphysics scheme with those using the Morrison double-moment bulk scheme employing saturation adjustment (Morrison et al. 2005). Rainfall accumulations over the simulated 6-hour period were about twice as
large in the bulk scheme as in the bin scheme, and there were significant differences in the surface rain accumulation maps (e.g., see Figs. 3, 4, and 5). These seem to agree with Khain and Lynn (2009) differences. Profiles of the mean convective core updrafts in Lebo and Seinfeld (2011) differ significantly as well, with stronger updrafts in the bulk scheme (see Figs. 7 and 13 there). Using the same framework and different modifications of the Morrison bulk scheme, Lebo et al. (2012) show values of the supersaturations in their simulations in excess of 10% (see Fig. 13 there). Simulations of shallow to deep convection
transition based on observations over the Amazon discussed in Grabowski and Morrison (2016, see Figs. 9 and 13 there; and 2020, see Fig. 10 there) also show that the supersaturations in deep convective cores below the freezing level can reach up to 10%, with several percent supersaturation differences between pristine and polluted conditions. Zhang et al. (2021) discuss simulations of deep convection over the Houston area applying the Morrison scheme as in Lebo and Seinfeld (2011) together with Khain et al. (2004) bin microphysics. They show similar convective organization in simulations applying the two schemes
and argue that bin results are in a better agreement with observations. Although the differences between results from simulations applying the two schemes are relatively small, there is some convective invigoration as represented by stronger updraft velocities and larger surface precipitation in the polluted case.

The purpose of the current study is to analyze contributions to the updraft buoyancy applying theory (section 3) and results of
numerical simulations (section 4). The impact of finite supersaturation on general aspects of convective dynamics (e.g., on CAPE) has not been investigated previously. Grabowski and Morrison (2016, 2020) argue that the impact of pollution on convective dynamics as simulated by a double-moment microphysics scheme predicting in-cloud supersaturation comes from the supersaturation differences between pristine and polluted conditions. This is because the impact is limited to the lower troposphere (i.e., below the melting level) and it is absent in simulations applying saturation adjustment (Grabowski 2015; see
also Grabowski and Morrison 2021). However, details of the buoyancy differences have not been discussed in detail, for instance, contributions from the entrainment to pristine-polluted differences and the impact of condensate loading. Theoretical analysis and additional analysis of simulations presented in our previous papers is the focus of the current manuscript.

**2 Cloud model simulations**


**2.1 The model setup and microphysics schemes**



Model simulations analyzed here were previously discussed in Grabowski (2015; G15 hereinafter), Grabowski and Morrison (2020; GM20), and Grabowski and Morrison (2021). A short description of the simulations is given below, with details

provided in the above publications.

G15 and GM20 apply a daytime convective development modeling case from Grabowski et al. (2006). The 12-hour long simulations, an extension of 6-hour simulations in Grabowski et al. (2006), start with the observed morning sounding and are driven by the surface sensible and latent heat fluxes. The fluxes strongly increase with the daytime surface insolation, reach a

mid-day maximum, and decrease to zero during the 10th hour of the simulation (see Fig. 1 in G15). The morning increase of the surface fluxes leads to the development of a well-mixed convective boundary layer, followed by the formation of shallow convective clouds, transition from shallow to deep convection around local noon, and only remnants of upper-tropospheric anvils present at the end of the simulations. Because convective development is forced only by prescribed surface heat fluxes, the same in all simulations, the differences in simulated convection come from different microphysics representations. The

simulations feature a doubly-periodic 50 km by 50 km horizontal domain with 400 m horizontal grid length. In the vertical, a stretched grid with 81 levels is used with about 10 (20) levels in the lowest 1 (4) km, reaching up to 24 km height. A small ensemble of simulations (see details below) is run for each case, with ensemble members generated by different sets of random numbers applied during the initialization and during model run as detailed in Grabowski et al. (2006).

G15 applied a relatively simple single-moment bulk microphysics scheme, referred to as IAB (Ice A and B, Grabowski 1999). IAB includes a simple warm-rain parameterization with a prescribed droplet concentration, 100 versus 1,000 per cc to mimic pristine (PRI) versus polluted (POL) conditions. The assumed droplet concentration affects conversion from cloud water to rain. IAB uses saturation adjustment to calculate cloud water condensation and evaporation. The ice parameterization is simple and is linked only indirectly to the assumed droplet concentration. Two classes of the ice mixing ratio are considered: slowly

falling ice A and fast-falling ice B. Ice A represents unrimed or lightly rimed ice particles whose spectral characteristics are assumed to follow aircraft observations in tropical upper-tropospheric anvil clouds. Ice B, on the other hand, represents heavily rimed ice particles (e.g., graupel) which occur in the vicinity of convective towers. Besides G15, the IAB scheme was successfully applied in deep convection simulations described in Varble et al. (2011), Fridlind et al. (2012), and Mrowiec et al. (2012). For more details, see the brief discussion in section 2b of G15 or a full description in Grabowski (1999).


The GM20 double-moment (2MOM) bulk microphysics scheme is more comprehensive. Droplet concentration is predicted together with the in-cloud supersaturation. CCN activation together with cloud droplet growth and evaporation are calculated explicitly from the predicted supersaturation instead of relying on saturation adjustment (Morrison and Grabowski 2007, 2008a). The warm-rain component predicts both number and mass mixing ratios for cloud water and rain (i.e., four Eulerian

variables). The double-moment three-variable ice microphysics of Morrison and Grabowski (2008b) predicts the number mixing ratio of ice particles and two mass mixing ratios that represent the ice mass grown by the diffusion of water vapor and





by riming. This allows a smooth transition from unrimed ice particles to heavily rimes particles (i.e., graupel) instead of artificially dividing ice particles into cloud ice, snow, and graupel categories that requires introduction of unphysical conversion rates. Primary ice initiation occurs through several processes, including deposition/condensation freezing,

heterogeneous freezing of cloud droplets and rain drops, contact freezing of cloud droplets, and homogeneous freezing of all droplets and drops for temperatures below -40°C. The key feature of the scheme is the close link between ice and warm-rain processes, the connection between droplet and ice number mixing ratios in particular. See section 2a in Grabowski and Morrison (2016) for a more extensive discussion or a full description of the scheme in Morrison and Grabowski (2007, 2008a,b). The simulations discussed in GM20 include a pristine case (PRIS) that features a single CCN mode with the number

mixing ratio of 100 mg$^{-1}$, and a polluted case with an additional mode of smaller CCN with 500 mg$^{-1}$ number mixing ratio, referred to as ADCN.

Results from the following ensemble simulations will be discussed. Four ensembles, two for IAB (PRI and POL) and two for 2MOM (PRIS and ADCN) relax the simulated mean horizontal winds to prescribed profiles as in Grabowski et al. (2006),

with four ensemble members for IAB's PRI and four for IAB's POL, and seven ensemble members for 2MOM's PRIS and seven for 2MOM's ADCN. Horizontal winds in Grabowski et al. (2006) feature significant shear (cf. Fig. 9 in GM20) and because of that we also consider the 2MOM no-horizontal-wind simulations in GM20 as in Wu et al. (2009) and Böing et al. (2012). These two three-member ensembles are referred to as PRIS.NW and ADCN.NW (NW for "no wind"). The reason for different numbers of ensemble members will become apparent in the discussion of model results (see Fig. 2).


## 2.2 Illustration of macroscopic cloud field characteristics

To highlight key similarities and differences between the IAB and 2MOM simulations, Figs. 1 and 2 show vertical cross section snapshots from randomly selected ensemble members at hour 6 (i.e., 360 min) and evolutions of the cloud cover and

total condensed water mass for the entire simulation length. The vertical cross sections in the X (east-west) and Y (north-south) directions are taken at the location of domain-maximum vertical velocity. At 360 min, the 2MOM simulation in Fig. 1 features a very deep convective tower that seems detached from the lower levels in the east-west and north-south cross sections, but this simply reflects the effects of shear causing the cloud to "lean" along the diagonal. There are also a few shallow clouds. In the IAB simulation snapshot, the cloud containing the domain-maximum vertical velocity at hour 6 is shallower but growing

rapidly. The cloud still has traces of lower-tropospheric moist static energy as shown by orange and red colors. The mean pattern of the moist static energy is similar between the two simulations. In Fig. 2, the cloud cover in left panels is defined as the fraction of columns with at least one grid volume featuring total condensate larger than 0.1 g kg$^{-1}$, with the total condensate including all cloud and precipitation mixing ratios. The right panels in Fig. 2 show evolution of the total condensate mass inside the computational domain. Total condensate increases due to the condensation and deposition, and decreases due to

evaporation, sublimation and precipitation reaching the surface. Figure 2 shows that there are small differences in IAB's PRI



and POL simulations as already discussed in G15. In contrast, the 2MOM simulations show large differences between PRIS and ADCN in the second half of the simulations. These differences are argued in GM20 (and also in Grabowski and Morrison 2016) to come from the microphysical impact of pristine versus polluted CCN conditions on the upper-tropospheric anvils. In a nutshell, higher cloud droplet concentrations in the polluted case result in higher ice crystal concentrations aloft, and this

leads to their smaller mean sizes, lower sedimentation velocities, and thus longer residence times. Figure 2 also illustrates different variability between ensemble members depending on the presence or absence of the mean large-scale flow, which informed our ensemble size selection. The vertical dotted lines in Fig. 2 mark the period of hours 6 and 7 (300 to 420 min) with the strongest deep convection that is used for the analyses presented in subsequent sections.

**3 Theoretical considerations: idealized parcel calculations**

We use domain-averaged temperature and water vapor mixing ratio profiles for the 6$^{th}$ and 7$^{th}$ simulation hours (i.e., averaged between 300 to 420 min and over all ensemble members) in a rising parcel analysis. The change of a generic thermodynamic quantity $\Phi$ with height for a rising parcel is given by (similar to Betts 1973, neglecting detrainment):

$$\mathrm{d}\Phi/\mathrm{d}z = -\varepsilon \, (\Phi - \Phi^e) + S_\Phi \qquad (2)$$


where $\varepsilon$ is the fractional entrainment rate, $S_\Phi$ is the source/sink of $\Phi$ owing to cooling by expansion and water phase changes, and $\Phi^e$ is the corresponding environmental value taken from the thermodynamic profile used in the analysis (i.e., the domain average potential temperature and water vapor for the 6$^{th}$ and 7$^{th}$ hour; the environment cloud water mixing ratio is assumed zero). Considering $\Phi$ as a moist conserved or nearly conserved quantity such as total water mixing ratio $q_t$ (water vapor plus

condensed water, neglecting removal by sedimentation) or equivalent potential temperature $\Theta_e$, the source/sink term $S_\Phi$ is (or is close to) zero. Using the base state pressure profile and the simplified form of $\Theta_e$ used for the analysis later (see section 4b), which is analogous to moist static energy, it is equivalent to solve (2) for $\Theta_e$ and $q_t$ and diagnose $\Theta$, water vapor mixing ratio $q_v$, and condensate mixing ratio $q$ needed for the buoyancy from $\Theta_e$ and $q_t$, *or* solve (2) directly for $\Theta$, $q_v$, and $q$. We chose the latter approach. As shown in section 4b, solving (2) assuming a constant $\varepsilon$ can reasonably well reproduce updraft $\Theta_e$ profiles

from the simulations. Thus, the simple parcel approach given by (2) can capture bulk behavior of the simulated updrafts.

The derived parcel $\Theta$, $q_v$, and $q$ are subsequently applied to obtain buoyancy profile using (1) and then to calculate the cumulative convective available potential energy (cCAPE) at height $z$ defined as:

$\mathrm{cCAPE}(z) = \int_0^z \max(0, B) \, dz \qquad (3)$



Buoyancy is calculated assuming an air parcel with the initial temperature and water vapor values taken as the mean in the lowest 500 m of the atmosphere and starting at the 500 m height. The mean low-level temperature and moisture values change little between the IAB and 2MOM ensembles (less than 0.1 K for the temperature and less than 0.1 g kg$^{-1}$ for the water vapor);

such changes have small impact on the results (e.g., ~100 J kg$^{-1}$ or less than 5% for the pseudo-adiabatic CAPE). Profiles from the 2MOM PRIS ensemble are used in the analysis presented below. The total CAPE is equal to cCAPE at the updraft equilibrium level, that is, at the level where updraft $B$ changes aloft from positive into negative.

The pseudo-adiabatic parcel analysis excludes the condensate term $q$ in (1); that is, the analysis assumes that the condensate is converted to precipitation and falls out with no impact on the parcel buoyancy. In the traditional parcel analysis, the rising

parcel is assumed to maintain water saturation at all heights above the lifting condensation level (LCL), in other words, applying saturation adjustment. However, one can assume nonvanishing supersaturation $S$, say, assuming $S$=0, 0.05, and 0.1 throughout the entire troposphere (i.e., 0, 5, and 10% supersaturation). For comparison, we also include parcel analysis with the buoyancy that includes a fraction of the condensate loading $f$ of 1/3, 2/3 and all condensate ($f$=1) in (1) at each level; thus, we implicitly assume a fraction 1-$f$ of the condensate is removed by conversion to precipitation followed by

sedimentation. Finally, we also consider the impact of parcel dilution assuming three different fractional entrainment rates, $\varepsilon$ = 0.05, 0.1, and 0.3 km$^{-1}$. This range of $\varepsilon$ is broadly consistent with bulk fractional entrainment rates derived from previous modeling studies for deep convection (e.g., Kuang and Bretherton 2006; Del Genio and Wu 2010; De Rooy et al. 2013) and derived from the simulations in section 4b. Assuming a classical entrainment formulation of $\varepsilon \sim 0.2/R$ where $R$ is the parcel radius (e.g., de Rooy et al. 2013), gives $R$ = 4, 2, and 0.6 km for the three $\varepsilon$ values selected.

Figure 3 shows profiles of cCAPE and buoyancy for the parcel analysis with numerical values at heights of 4 and 9 km presented in Table 1. For the pseudo-adiabatic parcel (Fig. 3a,b), limiting the parcel to water saturation ($S$ = 0) provides the largest buoyancy, at least in the lower and middle troposphere; consistent with the theoretical analysis and simulations of Grabowski and Jarecka (2015) and Grabowski and Morrison (2017). Retaining supersaturated conditions in the finite-supersaturation pseudo-adiabatic parcel leads to a small but noticeable reduction of buoyancy, cCAPE, and CAPE (see Table

1). It also allows additional latent heating above 11 km, but the upper-tropospheric buoyancy difference has little impact on the total CAPE because CAPE is dominated by the lower- and middle-tropospheric buoyancy differences. The pseudo-adiabatic buoyancy increases up to about 8-km height, and decreases in the upper troposphere, with the total CAPE reaching values around 2,500 J kg$^{-1}$. Calculating a theoretical updraft vertical velocity as $w = \sqrt{2\,cCAPE}$, obtained by vertically integrating the parcel vertical velocity equation neglecting perturbation pressure forcing and momentum mixing, gives values

around 25 m s$^{-1}$ for $S$=0 and around 21 m s$^{-1}$ for $S$=10% at 4 km. At 9 km, these values are around 53 m s$^{-1}$ for $S$=0 and around 49 m s$^{-1}$ for $S$=10%. These differences are relatively small, but nonnegligible. Updraft vertical velocities simulated by the dynamic model (see the next section) are two to three times smaller presumably because of entrainment, the missing loading





term in the pseudo-adiabatic parcel analysis, and excluding perturbation pressure forcing in the parcel model that all (in general) limit the theoretical updraft strength below its equilibrium level.


**Table 1: Results from parcel simulations. The table shows CAPE together with cCAPE and buoyancy at 4 and 9 km height. The first three rows show results from the pseudo-adiabatic parcel analysis assuming saturation adjustment (*S*=0) and *S*=5 and 10% throughout the atmosphere. The middle three rows show results from parcel calculations that assume saturation adjustment and**

**include 1/3, 2/3, and full condensate loading. The bottom three rows show results of entraining parcel calculations assuming saturation adjustment and no loading with three entrainment rates, 0.05, 0.1 and 0.3 per km. The initial temperature and water vapor values in the parcel are taken as averages in the lowest 500m and over hour 6 and 7 from the 2MOM PRIS ensemble.**

| | CAPE (J kg$^{-1}$) | cCAPE (J kg$^{-1}$) | | buoyancy (m s$^{-2}$) | |
|---|---|---|---|---|---|
| | | at 4 km | at 9km | at 4 km | at 9 km |
| no loading, no entrainment | | | | | |
| S=0 | 2572 | 321 | 1396 | 0.152 | 0.260 |
| S=5% | 2488 | 264 | 1301 | 0.138 | 0.255 |
| S=10% | 2407 | 220 | 1211 | 0.124 | 0.251 |
| | | | | | |
| S=0, loading, no entrainment | | | | | |
| f=1/3 | 2051 | 277 | 1182 | 0.131 | 0.212 |
| f=2/3 | 1553 | 243 | 967 | 0.110 | 0.164 |
| f=1 | 1076 | 208 | 752 | 0.088 | 0.116 |
| | | | | | |
| S=0, no loading, entrainment | | | | | |
| ε=0.05 km$^{-1}$ | 1562 | 288 | 1118 | 0.134 | 0.167 |
| ε=0.1 km$^{-1}$ | 1024 | 266 | 896 | 0.119 | 0.100 |
| ε=0.3 km$^{-1}$ | 380 | 194 | 380 | 0.071 | 0 |


Including condensate loading in the parcel buoyancy has a significant impact. Using all condensate in the parcel buoyancy is arguably appropriate just above the cloud base before the condensate is reduced by precipitation fallout. However, it is questionable in the middle and upper troposphere because it implies over 10 g kg$^{-1}$ of cloud condensate, a clearly unrealistic

value (this will be illustrated by the analysis later in the paper). Nevertheless, even a third of the cloud condensate notably reduces parcel buoyancy, cCAPE and CAPE. The theoretical updraft strength at 4 km changes from about 24 m s$^{-1}$ for *f*=1/3





to about 20 m s$^{-1}$ for $f$=1; at 9 km these values are 49 m s$^{-1}$ and about 39 m s$^{-1}$, respectively. Entrainment has a large impact as well, with the smallest fractional entrainment rate tested (0.05 km$^{-1}$) reducing cCAPE by about 10% at 4 km and 20% and 9 km compared to an undilute parcel, and the total CAPE by about 25%. The theoretical updraft $w$ at 4 km changes from about 25 m s$^{-1}$ for $\varepsilon$=0.05 km$^{-1}$ to about 20 m s$^{-1}$ for $\varepsilon$=0.3 km$^{-1}$, similar to the impact of including condensate loading. At 9 km these values for $\varepsilon$ of 0.05 and 0.3 km$^{-1}$ are 47 m s$^{-1}$ and 39 m s$^{-1}$, respectively. The largest fractional entrainment rate tested, $\varepsilon$=0.3 km$^{-1}$, gives an equilibrium level in the mid-troposphere (~8 km) compared to > 12 km in the other tests (Fig. 3f). The lower equilibrium height with greater $\varepsilon$ is expected and is consistent with previous theoretical (Morrison et al. 2021) and cloud modeling (e.g., Kuang and Bretherton 2006) studies. Finally, we point out that there is almost no impact of entrainment or loading on *absolute* differences in cCAPE resulting from changes in $S$ between 0 and 10% (not shown). This can be understood by the fact that changes in parcel temperature from entrainment are small relative to the parcel temperature itself (~1% or less). Thus, changes in parcel temperature owing to changes in $S$ can be well approximated as being independent of entrainment. Because the magnitude of cCAPE decreases from entrainment, the *relative* change in cCAPE from finite $S$ increases with greater entrainment. Similarly, changes in $q$ owing to finite $S$ (up to 10%) are small relative to $q$ itself. Thus, including loading has little impact on absolute changes to cCAPE from finite $S$, although relative changes to cCAPE increase.

In summary, the theoretical analysis presented in this section shows that finite supersaturation (up to 10%) has a nonnegligible impact on convective dynamics. However, the impact is relatively small overall when compared to the effects of condensate loading and entrainment. Moreover, including loading or entrainment has almost no impact on absolute differences in cCAPE resulting from changes in $S$ (but relative differences in cCAPE increase since both entrainment and loading act to decrease cCAPE). Larger changes in cCAPE and $w$ might be possible if $\varepsilon$ or $f$ are different in polluted compared to pristine conditions. These factors are considered further in dynamic model simulations discussed in the next section.

## 4 Results of dynamic simulations

### 4.1 Entrainment and its impact on updraft velocity

To characterize the impact of entrainment on the buoyancy and updraft velocity, we apply an equivalent potential temperature $\Theta_e$ defined here as the moist static energy divided by $c_p$:

$$\Theta_e = T + g/c_p \; z + L_v /c_p \; q_v \qquad (4),$$

where $T$, $z$, and $q_v$ are the temperature, height, and water vapor mixing ratio, and $L_v$ and $c_p$ are the latent heat of condensation and air specific heat at constant pressure. For a rising adiabatic or pseudo-adiabatic parcel with no ice processes, the equivalent potential temperature $\Theta_e$ defined in (4) is an *invariant* for the anelastic model and the moist precipitating thermodynamics



applied in both saturation adjustment (IAB) and saturation prediction (2MOM) ensembles. Moreover, regardless of the condensate amount carried by the cloudy air, mixing between the cloudy air parcel and subsaturated cloud-free environmental parcel results in the equivalent potential temperature that is a linear combination of the relative mass contributions of the two parcels $\Theta_e$.


The left panel in Figure 4 shows the equivalent potential temperature $\Theta_e$ statistics for in-cloud points with updraft velocity larger than 1 m s$^{-1}$ and total condensate larger than 1 g kg$^{-1}$ for all members of GM20's PRIS and ADCN ensembles during 6$^{th}$ and 7$^{th}$ simulation hours. The dashed line shows the $\Theta_e$ profile of the initial sounding and solid lines show $\Theta_e$ profiles of the mean (domain- and time-averaged) temperature and moisture during hours 6 and 7. The difference between the dashed and
solid lines represents the impact of surface latent and sensible heat fluxes combined with the vertical transport throughout the column. Arguably, radiative cooling during the day and especially throughout the night, not considered in the simulations, would be needed to bring the dashed and solid lines closer to each other if the simulations were extended to several diurnal cycles (i.e., approaching convective-radiative quasi-equilibrium).

Undiluted ascent from the cloud base would correspond to a vertical line in Fig. 4 left panel, so there is almost always some cloud dilution by entrainment of environmental air. The figure is similar to higher-resolution simulations of this case in Khairoutdinov and Randall (2006); see Figs. 11 and 12 there. This is in contrast to results presented in Varble et al. (2014, see Fig. 16 there) that show very little dilution across the entire troposphere for an ensemble of mesoscale convective system simulations, possibly because of the organized nature of convection in their simulations compared to the unorganized
convection here and in Khairoutdinov and Randall (2006). Dilution corresponding to the mean $\Theta_e$ as well as the minimum dilution (i.e., the mean of the 90$^{th}$ percentile to the maximum range of $\Theta_e$) increase with height as expected; that is, the deviation from the cloud-base $\Theta_e$ increases as one moves away from the cloud base. There seems to be a small impact of the microphysics on entrainment dynamics between 3 and 5 km, with slightly smaller mean values of $\Theta_e$ for ADCN compared to PRIS. Such a difference may come from small but systematic differences in the mixing proportions between cloud and environmental air in
ADCN and PRIS. Plots for the no-wind GM20 ensembles (PRIS.NW and ADCN.NW) are similar, including the small difference between PRIS.NW and ADCN.NW (not shown). Plots for IAB are also similar, although there is no difference between PRI and POL, arguably because saturation adjustment limits differences in latent heating and cooling that impact entrainment dynamics. The latter is consistent with high resolution simulations in Grabowski and Jarecka (2015). Finally, a kink of the mean and median values (better seen in the right panel discussed below) near 5 km likely comes from $\Theta_e$ changes
due to ice processes, particularly melting.

Linking directly to the theoretical analysis in section 3, we use profiles of updraft $\Theta_e$ from the simulations to estimate constant-in-height fractional entrainment rate $\varepsilon$ using our entraining parcel model. This is done by vertically integrating (2) for $\Phi = \Theta_e$ between about 1 km (near the level of free convection) and 10 km, using $\Theta_e$ from the simulations as a lower boundary condition.





$\varepsilon$ is estimated by finding the value that minimizes the root-mean-square difference between profiles of $\Theta_e$ obtained by solving (2) and from the simulations. Note that this simple approach estimates a *bulk* $\varepsilon$ value obtained by assuming entrained properties of air are equal to those of the average environment for hours 6 and 7. This is different from *direct* entrainment calculations based on the mass fluxes across cloud updraft boundaries (e.g., Romps 2010, Dawe and Austin 2013). Direct calculations of $\varepsilon$ are generally larger than bulk estimates by up to about a factor of 2 (Romps 2010). For simplicity, and because we are

concerned primarily with updraft dilution as opposed to entrainment per se, we use the bulk approach. Two different $\Theta_e$ profiles from the simulations are used to estimate $\varepsilon$: the median and 90th percentile $\Theta_e$. This gives $\varepsilon$ for "typical" updrafts as well as relatively undilute updrafts (or updraft regions). The best fit $\varepsilon$ are ~0.20 km$^{-1}$ for median $\Theta_e$ and 0.13 km$^{-1}$ for 90th percentile $\Theta_e$. These are similar to previous bulk estimates for deep convection (e.g., Kuang and Bretherton 2006; Del Genio and Wu 2010; De Rooy et al. 2013). The right panel of Fig. 4 compares the median and 90th percentile of the dynamic model $\Theta_e$

distributions with entraining parcel calculations assuming those best fits. In general, the $\Theta_e$ profiles from solving (2) with these $\varepsilon$ values reasonably reproduce the simulated profiles with root-mean-square differences of ~0.8 and 0.4 K for median and 90th percentile $\Theta_e$, respectively. Larger differences in the upper troposphere can be explained by a smaller entrainment rate that allows those cloudy volumes to be present there in the first place (see bottom panel in Fig.1), as well as the impact of ice microphysics on $\Theta_e$ which is neglected in the parcel calculations. A key result is that best-fit $\varepsilon$ values are almost the same for

pristine and polluted simulations: 0.203 and 0.206 km$^{-1}$ for PRIS and ADCN for median $\Theta_e$ and 0.131 and 0.127 km$^{-1}$ for PRIS and ADCN for 90th percentile $\Theta_e$. Thus, in these simulations microphysical differences from polluted versus pristine conditions appear to have little impact on overall entrainment behavior. The small differences in $\Theta_e$ between ADCN and PRIS seen in Fig. 4 at mid-levels (~3 to 5 km) evidently have little impact on bulk entrainment differences considering the entire profile between 1 and 10 km.


Figure 5 shows scatterplots of the equivalent potential temperature versus vertical air velocity at a height of 4 km during hours 6 and 7 for all IAB and 2MOM ensemble members and for grid volumes with $w$ larger than 1 m s$^{-1}$ and total condensate larger than 1 g kg$^{-1}$. On average, updraft strength increases with $\Theta_e$, but there is a significant scatter. The highest $\Theta_e$ for all ensembles is close to the cloud base $\Theta_e$ (~250 K as shown in Fig. 4). The saturation adjustment simulations using IAB, PRI and POL,

feature the strongest updrafts and there is no difference between them except for different flow realizations. In contrast, ADCN and ADCN.NW have stronger updrafts when compared to PRIS and PRIS.NW, but the differences are relatively small, most evident in the difference in the number of points in the range of 5 to 10 m/s. For that range, the mean updraft in IAB ensembles is 6.87 m s$^{-1}$ for PRI and 6.91 m s$^{-1}$ for POL, arguably a statistically insignificant difference. For 2MOM ensembles, the differences are larger, 6.26/6.53 m s$^{-1}$ for PRIS/ADCN and 6.25/6.62 m s$^{-1}$ for PRIS.NW/ADCN.NW.


Figure 6 shows similar results as Fig. 5, but at 9 km height and only for PRIS and ADCN ensembles. Compared to Fig. 5, updrafts are stronger than at 4 km (consistent with higher buoyancy and the increase of cCAPE with height, see Fig. 1), the



equivalent potential temperature maxima are lower (i.e., more dilution), and there is less scatter. The difference between PRIS and ADCN seems to be absent (the latter is also true for IAB ensembles and no-wind 2MOM ensembles; not shown). The

mean updraft for the 5 to 10 m s⁻¹ range is 7.06/7.08 m s⁻¹ for 2MOM's PRIS/ADCN and 7.13/7.11 m s⁻¹ for IAB PRI/POL, that is, only slightly larger for the saturation adjustment IAB simulations. The latter is not surprising as ice microphysics takes over at that height as illustrated in Fig. 7, to be discussed shortly. For the 10 to 20 m s⁻¹ range, the mean values are 12.1 m s⁻¹ for both PRIS and ADCN from 2MOM ensembles and 13.1/12.9 m s⁻¹ for PRI/POL from IAB.

**4.2 Supersaturation, buoyancy and updraft statistics**

Figures 7 shows profiles of the supersaturation, updraft, and buoyancy statistics for rising (updraft larger than 1 m s⁻¹) cloudy (total condensate larger than 1 g kg⁻¹) volumes for ensemble members during hours 6 and 7 from IAB and 2MOM (PRIS and ADCN only for the latter). The overall impression is that the statistics look fairly similar between the two figures with the

exception of the supersaturation. The supersaturation statistics for IAB show that the model's saturation adjustment works correctly in the lower and middle troposphere. Only above 9 km, ice processes allow supersaturation in ascending cloudy volumes to become subsaturated with respect to liquid water resulting in a range of subsaturation values. The range of subsaturation above 9 km seems similar between IAB and 2MOM ensembles. For 2MOM, supersaturations can be significant (several percent), with the mean values (slightly lower for ADCN) ranging from 1 to 5% below the freezing level. The mean

of the 90th percentile to the maximum range in Fig. 8 increases with height and reaches values close to 10/15% in the middle troposphere for ADCN/PRIS ensembles.

Large supersaturation differences between IAB and 2MOM ensembles and differences between PRIS and ADCN lead to noticeable differences in the buoyancy statistics, in agreement with the discussion in Grabowski and Jarecka (2015) and

Grabowski and Morrison (2017). In the lower and middle troposphere, buoyancies are significantly larger in IAB ensembles when compared to 2MOM, and slightly larger in ADCN when compared to PRIS. The mean and the maximum buoyancies increase with height below the melting level in all ensembles (in agreement with the parcel analysis, Fig. 1), reach maximum values near the melting level, and then level off. Although the buoyancies do include all terms shown in Eq. (1), the maxima near the melting level are only slightly smaller than the values predicted by the adiabatic parcel, that is, around 0.15 and 0.13

m s⁻² for $S$=0 and $S$=10%, respectively, at 4 km in Table 1.

Despite the differences in the supersaturation and buoyancy, overall updraft statistics are similar. Maximum updraft vertical velocities increase with height in agreement with the parcel analysis (Fig. 1) and buoyancy statistics in the cloud model simulations. Below the freezing level, updrafts are stronger in IAB than in 2MOM, especially at the maximum end. At the

freezing level, the mean updraft values of the 90th percentile to the maximum range are around 11 m s⁻¹ in IAB ensembles and around 7 and 8 m s⁻¹ in PRIS and ADCN, respectively. Above the 0° C level, the mean updraft values of the 90th percentile to



the maximum range increase with height similarly between all ensembles, and they reach 15 to 20 m s$^{-1}$ in the upper troposphere. Results for the no-wind 2MOM ensembles (PRIS.NW and ADCN.NW) are similar to those shown in Fig. 7 and thus are not shown.


**4.3 Buoyancy analysis**

To understand buoyancy differences better in 2MOM simulations and the lack thereof in IAB, Figures 8 to 11 show contributions of buoyancy components as a function of the total buoyancy at 4 and 9 km for the IAB ensembles (Figs. 8 and

9) and 2MOM ensembles (Figs. 10 and 11) for grid volumes with updraft velocities larger than 1 m s$^{-1}$ and total condensate (cloud plus precipitation) larger than 1 g kg$^{-1}$. Buoyancy components represent the three terms in (1): the temperature, virtual, and loading components. In addition, the loading component is split into cloud and precipitation contributions (e.g., cloud water and rain at 4 km) as discussed below. Overall, the four figures show a coherent picture of the buoyancy contributions, with only small differences between the four ensembles. In agreement with the discussion above, IAB ensembles reach larger

buoyancy values (note different buoyancy ranges on the horizontal axes in Figs. 8/9 and 10/11). In all four ensembles, the temperature term (i.e., the latent heating) provides the largest contribution. The temperature contribution increases linearly with the total buoyancy, with some scatter.  The temperature contribution is aided by water vapor (i.e., the virtual temperature effect, especially at 4 km), and offset by the loading. For IAB at 4 km (Fig. 8), the loading includes only cloud water and rain, but at 9 km in Fig. 9 cloud liquid and ice A are merged together to represent "cloud condensate", and rain and ice B are

combined as "precipitation". There are almost no differences between left and right panels in Figs. 8 and 9, consistent with saturation adjustment, just different flow realizations. The largest buoyancies are in volumes with relatively small contributions from the loading (especially at 9 km, Fig. 9) and the maximum buoyancies are not far from the pseudo-adiabatic parcel analysis values shown in Table 1 that excludes the loading.

For the 2MOM ensembles (Fig. 10 and 11), the maximum buoyancies are smaller than for IAB, but the patterns are similar. For the loading at 9 km (Fig. 11), the ice mass mixing ratio grown by diffusion of water vapor ($q_{id}$) is included in the "cloud condensate" contribution, and the ice mass mixing ratio grown by riming ($q_{ir}$) in the "precipitation condensate". Although there are small differences between PRIS and ADCN in total buoyancy as seen in Fig. 8, mainly below 5 km, these differences are not readily apparent in Figs. 10-11 when partitioned into the various contributions.


In the following analysis, we focus on the 2MOM simulations to better understand the buoyancy differences in PRIS and ADCN ensembles. We take advantage of the piggybacking technique that was used in G15, Grabowski and Morrison (2016, 2017) and in GM20. Piggybacking applies two sets of thermodynamic variables (the temperature, water vapor, and all aerosol, cloud, and precipitation variables) in a single cloud field simulation. The first set is coupled to the dynamics and

drives the simulation; hence the driver. The second set, the piggybacker, is carried by the simulated flow and is modified by





the same physical processes as the driver (e.g., surface fluxes, latent heating, precipitation fallout, etc.), but it does not affect the flow (see Grabowski 2019 for the discussion and examples of application). Because every grid volume features two sets of cloud and thermodynamic variables (i.e., from the driver and from the piggybacker), these variables can be directly compared grid point by grid point instead of using conditional sampling. As a result, piggybacking allows one to separate the

impact of a physical process (e.g., diffusional growth of cloud droplets or conversion from cloud water to rain) from the effects of different flow realizations.

Figure 12 shows scatterplots of the driver versus piggybacker buoyancies for the PRIS and ADCN ensembles at 4 and 9 km during 6th and 7th hours of the simulations with colors indicating the updraft velocity. The same dataset (although only for

hour 6 and without colors) was used to show similar scatterplots at 3 and 7 km in Fig. 5 of GM20. At 4 km, ADCN buoyancies are larger than for PRIS (with a few exceptions) regardless if ADCN drives (lower right panel) or piggybacks (lower left panel) the simulation. The larger buoyancies correspond to larger vertical velocities (i.e., red and blue symbols), but the scatter is significant, perhaps because the local updraft magnitude represents the time-integrated buoyancy, not the current location value (as well as integrated impacts of perturbation pressure forcing). The differences at 4 km are relatively

small, typically below 0.02 m s$^{-2}$, especially when comparing the magnitude of extreme buoyancies shown in Fig. 10. Only a small fraction of points shows larger differences, especially in the 0 to 0.05 m s$^{-2}$ range. Red points (updrafts in 5 to 10 m s$^{-1}$ range) typically show larger buoyancies for the ADCN ensemble (consistent with the supersaturation differences), whereas blue points (updrafts above 10 m s$^{-1}$) are scattered both above and below the 1:1 line, showing contrasting impacts of the temperature and loading contributions (increasing/reducing buoyancy for the former/latter in the ADCN ensemble as shown

in Fig. 14 below). A combination of the temperature and loading terms arguably explains the small differences in the largest updraft velocities between PRIS and ADCN as shown in Figs. 5 and 7. Driver-piggybacker differences at 9 km are smaller, with ADCN buoyancies typically slightly larger regardless if ADCN drives or piggybacks the flow. In contrast to the 4-km plots, a clear trend of larger driver-piggybacker differences for larger buoyancies is evident. In addition, the strongest updrafts tend to correspond to the largest buoyancies, also in some contrast to the 4-km statistics.


Figures 13 and 14 show scatterplots in a similar format as Fig. 12 but contrasting buoyancy components between PRIS and ADCN ensembles at 4 km (Fig. 13) and 9 km (Fig. 14). The figures show buoyancy components for the temperature (a,b), water vapor (c,d), and condensate (e,f,g,h), the latter showing total condensate as well as cloud and rain water separately in Fig. 13 with ice components added in Fig. 14. Upper and lower dashed lines in the panels correspond to approximately the

same impact of the perturbations on the buoyancy as dashed lines above and below 1:1 line in Fig. 12 (i.e., around 0.02 m s$^{-2}$). At 4 km (Fig. 13), the temperature and condensed water (cloud water and rain) differences are the largest contributors to the PRIS-ADCN buoyancy differences. The water vapor PRIS-ADCN difference adds little; even a 10% difference in supersaturation and hence water vapor mixing ratio has little impact on buoyancy because the virtual temperature effect on buoyancy is already relatively small (see Fig. 10). The potential temperature is typically larger in ADCN than in PRIS



regardless of whether it drives or piggybacks the simulation. One can argue (e.g., Fan et al. 2018) that this is consistent with smaller supersaturations and larger condensational growth in ADCN. However, since the driver and piggybacker experience the same updraft, the temperature difference has to come from *the supersaturation being different from the quasi-equilibrium supersaturation*. This is because, as mentioned in the introduction and discussed in detail in section 2b of GM20, the condensation rate (and thus the latent heating) for a given updraft is the same as long as the supersaturation is equal to the

quasi-equilibrium supersaturation. In other words, for condensational growth, the driver and piggybacker temperatures would be the same if the supersaturation was equal to the quasi-equilibrium supersaturation. Another possibility is that the difference comes from entrainment, but this is unlikely as illustrated by Fig. 4 and its discussion.

The bottom panels in Fig. 13 show that cloud and rain water also contribute significantly to the buoyancy differences, aiding
the temperature differences seen in panel (a,b). Note that the analysis includes only points with the total condensate larger that $1$ g kg$^{-1}$, and this explains why there are no points with total condensate smaller than that threshold in (e,f). In panels (g,h), the red symbols for cloud water in the lower left panel are above the 1:1 line which implies that ADCN has more cloud water, in agreement with the suppressed conversion from cloud water to rain in polluted conditions. (Note that these points have to come from volumes with rain because – to be included in the plot - the total condensate has to be larger than $1$ g kg$^{-1}$). As a result,
the cloud water opposes the temperature ADCN-PRIS difference.  But this is counterbalanced by the rain contribution that helps the temperature impact. Points for the D-PRIS versus P-ADCN rainwater in the lower left panel (blue symbols) are below the 1:1 line. This implies that there is more rain water in the D-PRIS, again consistent with suppressed conversion of cloud water to rain in ADCN. At the same time, rain falls from above and also contributes negatively to the buoyancy at 4 km in D-PRIS. There is a significant scatter in the rain points and there are some points where rain is higher in ADCN, no doubt from
the impact of rain sedimentation from higher levels in some grid volumes. The right panels are close to mirror images of those in the left column. In summary, both the temperature difference (warmer in ADCN) and the loading difference (on average smaller in ADCN) contribute to the larger buoyancy in ADCN below the freezing level. However, the overall difference is small as shown in Fig. 11. An important point is that the temperature and loading differences between PRIS and ADCN results can only be seen in the comparison applying piggybacking because they are not seen in Fig. 10 buoyancy analysis.


Figure 14 shows analogous results at 9 km. For the temperature, the outcome is similar to that at 4 km (Fig. 13), with ADCN being slightly warmer compared to PRIS, especially at the highest temperature end. Contributions from water vapor are even smaller than in Fig. 13, as expected, and this is why panels (c,d) do not show dashed lines (these are outside the range shown on the axes). For the cloud and precipitation condensate, the total impact (panels e,f) is small with some scatter around the 1:1
line. Panels (g,h) show the contributions from the cloud water and the two ice mixing ratios. The rain water is close to zero and thus it is not shown. The cloud water range is similar to that at 4 km (up to ~1 g kg$^{-1}$) and it contributes negatively to the ADCN-PRIS difference. The ice mixing ratio grown by diffusion of water vapor ($q_{id}$, black symbols) shows small ADCN-PRIS differences except for large mixing ratios where it is larger in ADCN than PRIS ensembles in both the left and right





panels. One possibility is that the higher ice crystal concentrations resulting from higher droplet concentrations at lower levels
in the ADCN ensemble leads to a more efficient growth by the diffusion of water vapor, similarly to the condensational growth
of cloud droplets. The blue symbols for the ice mixing ratio grown by riming ($q_{ir}$) show a similar impact as that for rain at 4
km (i.e., below/above the 1:1 line in the left/right panels). This may come from frozen rain drops carried from lower levels
(more abundant in the PRIS case) together with more efficient growth by riming for larger cloud droplets in the PRIS case.
Comparing panels (e,f) and (g,h) clearly shows that there is a significant compensation between the mixing ratios of ice grown
by vapor deposition and riming that together lead to a relatively small impact on total condensate differences between PRIS
and ADCN.

In summary, the temperature and loading differences in the 2MOM simulations are fairly small and result in limited differences
in the buoyancy and vertical velocity in PRIS and ADCN ensembles. The buoyancy and vertical velocities in 2MOM ensembles
are smaller than in IAB because the saturation adjustment in IAB ensembles is replaced by the supersaturation prediction in
2MOM.

**5 Conclusions**

This paper investigates factors affecting cloud buoyancy using theoretical analysis and results from numerical simulations of
unorganized deep convection. The motivation comes from previous discussions of the so-called convection invigoration in
polluted environments, that is, the increase of updraft speed resulting from the increase of the CCN concentrations. A recent
exchange between Fan and Khain (2021) and Grabowski and Morrison (2021), and references in those papers, provide the
context for the invigoration conundrum. The original claim (e.g., Andreae et al. 2004, Rosenfeld et al. 2008) hypothesized that
suppression of rain formation in the lower troposphere in high-CCN environments, transport of the cloud water across the
melting level, and freezing it aloft results in the invigoration of the upper-tropospheric updrafts. However, as discussed in
Grabowski and Morrison (2020, section 2a) and mentioned in the introduction, the latent heat of freezing merely compensates
the loss of buoyancy due to carrying the extra liquid into the upper troposphere. Hence, the so-called "cold -phase invigoration"
is difficult to justify on theoretical grounds. Fan et al. (2018) argues that the presence of lower supersaturations in polluted
convective updrafts below the freezing level provides a different kind of invigoration that indeed has been seen in previous
simulations (Grabowski and Jarecka 2015, and Grabowski and Morrison 2017, 2020, see also Cotton and Walko 2021). This
has been referred to as "warm-phase invigoration" because the argument involves differences in the condensational growth of
cloud droplets below the freezing level. However, physical mechanisms behind the warm-phase invigoration are unclear,
especially when considered together with other processes affecting cloud buoyancy, such as the condensate loading,
precipitation fallout, and entrainment.



We analyze a 2-hour period of deep convection from 12-hour-long simulations of daytime convection development over land. The mean sounding from this period of the simulations serves as input to the theoretical analysis applying a rising parcel framework (see Fig. 3 and Table 1). For the pseudo-adiabatic parcel, that is, excluding parcel condensate loading, we contrast
results obtained with different levels of the supersaturation maintained within the parcel, from 0% (i.e., the traditional water-saturated parcel analysis) to 10% supersaturation (with respect to water saturation) across the entire depth the parcel rises. In agreement with previous theoretical analysis in Grabowski and Jarecka (2015), maintaining finite supersaturations results in a reduction of pseudo-adiabatic parcel buoyancy in the lower and middle troposphere, a several-percent reduction of the total CAPE, and a fairly large reduction of the cumulative CAPE in the lower troposphere (e.g., from about 300 to about 200 J kg$^{-1}$
at 4 km for 10% supersaturation). Including loading in the parcel buoyancy, relative to the pseudo-adiabatic parcel with no loading, has a similar impact in the lower troposphere as going from 0 to 10% supersaturation (e.g., similar cCAPE reduction at 4 km), but it is much more significant in the upper troposphere. This occurs because the supersaturation impact on parcel buoyancy decreases with height, but the condensate carried by the adiabatic parcel increases, approaching near-surface water vapor mixing ratios, over 15 g kg$^{-1}$, in the upper troposphere. For an adiabatic parcel with all condensate included in the parcel
buoyancy, the total CAPE is about 40% of the pseudo-adiabatic CAPE. In agreement with numerous past observational and modeling studies of deep convection, the impact of entrainment on parcel buoyancy and cCAPE is large, with a relatively small bulk fractional entrainment rate of 0.05 km$^{-1}$ reducing total CAPE by about 40%. Larger entrainment rates lower the equilibrium level of the parcel from the upper to the middle troposphere and substantially reduce cCAPE. The magnitude of cCAPE changes from finite supersaturation is not affected by condensate loading or entrainment. However, because loading
and entrainment reduce cCAPE the *relative* impacts of finite supersaturation increase.

The impact of the supersaturation, entrainment and loading is further quantified in the analysis of the numerical simulations. The simulations use either a simple single-moment bulk scheme with saturation adjustment, the IAB set of simulations, or more comprehensive double-moment bulk scheme with supersaturation prediction, the 2MOM simulation set. The difference
between pristine and polluted CCN conditions is simulated by assumed contrasting cloud droplet concentrations in the IAB simulations (Grabowski 2015; G15) or by specifying contrasting CCN spectra in the 2MOM simulations (Grabowski and Morrison 2020; GM20). For analyzing entrainment, we use a simplified formulation of equivalent potential temperature $\Theta_e$ equal to the moist static energy divided by $c_p$, a conserved variable for ice-free conditions and only slightly modified when ice is present (the latter is because the latent heat of freezing is only a small fraction of the latent heat of condensation). Profiles
of cloud updraft $\Theta_e$ statistics (Fig. 4) document a significant dilution by the entrained environmental air, with small differences between pristine and polluted conditions. Profiles of the median $\Theta_e$ correspond to a bulk fractional entrainment rate of about 0.20 km$^{-1}$, whereas profiles of the 90th percentile can be explained by a bulk fractional entrainment rate around 0.13 km$^{-1}$. These values are consistent with the idealized parcel simulations and vary insignificantly between 2MOM pristine and polluted simulations. The strongest updrafts, slightly stronger in the saturation-adjustment IAB simulations (see Fig. 5) occur in the
least diluted cloudy volumes, with small differences between pristine and polluted 2MOM simulations. The impact of





entrainment in the theoretical analysis and in model simulations based on median updraft properties versus those of strongest updraft cores is reminiscent of the old cloud physics problem of representing convective cloud properties using one-dimensional models (e.g., Warner 1970): a large entrainment is needed to represent overall cloud dilution, while at the same time only small entrainment rates ensure that the cloud depth (controlled by relatively less dilute parcels) is correctly

represented.

For the impact of finite supersaturation, we compare profiles of supersaturation, buoyancy, and updraft statistics between IAB simulations featuring saturation adjustment with 2MOM simulations that predict the in-cloud supersaturation. Finite supersaturations indeed provide a noticeable reduction of the updraft buoyancies and vertical velocities in the lower and middle

troposphere, with small differences between pristine and polluted conditions below and around the freezing level (Fig. 7). Analysis of in-cloud updraft buoyancies (Figs. 8 – 11) documents that the temperature term is the most significant buoyancy component (in agreement with the elementary arguments in the introduction), and it is opposed by the cloud and precipitation loading, especially below the freezing level in both IAB and 2MOM simulations (Figs. 8 and 10). Aloft, the largest buoyancies typically correspond to small loading; see Figs. 9 and 11. Overall, buoyancy contributions in IAB and 2MOM simulations are

similar, except for somewhat larger buoyancy maxima below the freezing level as mentioned above.

Because these simulations apply the piggybacking technique (i.e., each simulation carries two set of thermodynamic variables, one driving and one piggybacking the simulated flow), the impact of assumed CCN conditions on the buoyancy can be directly (i.e., point-by-point) compared between pristine and polluted conditions. The difference in the lower troposphere comes from

concurring differences in the temperature (due to latent heating) and loading. Because the driver and piggybacker experience the same cloud flow, the temperature difference must be explained by either the supersaturation being different from its quasi-equilibrium value or entrainment and mixing of temperature being different. The quasi-equilibrium supersaturation represents a balance between the supersaturation sink due to droplet diffusional growth and supersaturation source due to rising air motion. The key point is that the condensation rate is independent of the droplet concentration and size, and it depends only

on the vertical velocity, as long as the supersaturation is equal to it quasi-equilibrium value; see section 2b in GM20. Because our entrainment analysis shows similar behavior for pristine and polluted conditions (see section 4.1), the temperature difference can only be explained by the updraft supersaturations being different from their quasi-equilibrium values. At 4 km height (Fig. 13), slightly higher temperatures in the polluted case when compared to the pristine case are aided by the loading, larger in the pristine case, in agreement with suppressed rain formation in the polluted case. Temperature and loading

differences at 9 km height are smaller (Fig. 14), with an intriguing compensation between loading contributions from the ice mass grown by diffusion and that grown by riming.

Overall, the analysis presented in the paper suggests that the impact of CCN characteristics on convective dynamics is rather subtle and requires detailed analysis (e.g., through piggybacking) to understand the physical processes involved. Perhaps the





most significant difference from the modeling point of view is a contrast between the strongest updrafts when applying a microphysical scheme with saturation adjustment and one with supersaturation prediction, stronger in the former case (Grabowski and Morrison 2017, Zhang et al. 2021). In 2MOM simulations applying a scheme that predicts the in-cloud supersaturation, the differences between pristine and polluted conditions are rather small. How the latter depends on the particular microphysical scheme and whether it changes if a more sophisticated microphysics scheme is used (e.g., bin

microphysics as in Zhang et al. 2021 or Lagrangian microphysics as in Shima et al. 2020) needs to be investigated in the future.

*Data availability*. Grabowski, Wojciech W.: Buoyancy in Deep Convection Simulations. Version 1.0. UCAR/NCAR - DASH Repository, https://dashrepo.ucar.edu/dataset/206_grabow.html. DOI https://doi.org/10.5065/hqt3-1h72. Accessed 21 May 2021.


*Author contributions*. WWG ran cloud model simulations. HM run parcel simulations. WWG and HM performed data analysis and prepared the manuscript.

*Competing interests*. The authors declare that they have no conflict of interest.


*Acknowledgments.* Comments on a draft manuscript by NCAR's Andreas Prein are acknowledged. NCAR is sponsored by the National Science Foundation.

*Financial support*. This research has been partially supported by the U.S. DOE ASR Grant DE-SC0016476.

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

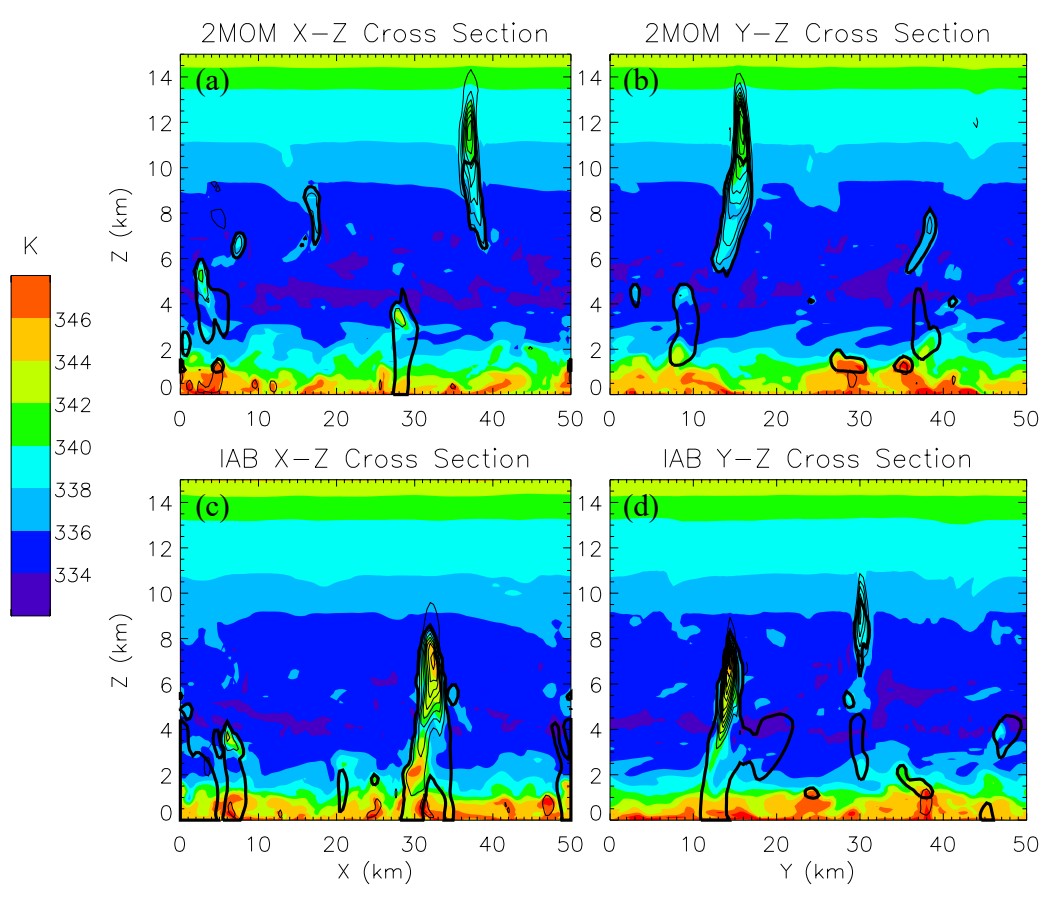

**Figure 1. Cross sections (x-z in a and c; y-z in b and d) though cloud simulations at 360 min from (a, b) 2MOM and (c, d) IAB randomly selected ensemble members. Colors represent the equivalent potential temperature. The thick contour shows total condensate (cloud and precipitation) of 0.1 g kg$^{-1}$. Sold thin contours show vertical velocity starting with 1 m s$^{-1}$ and contour interval of 3 m s$^{-1}$.**








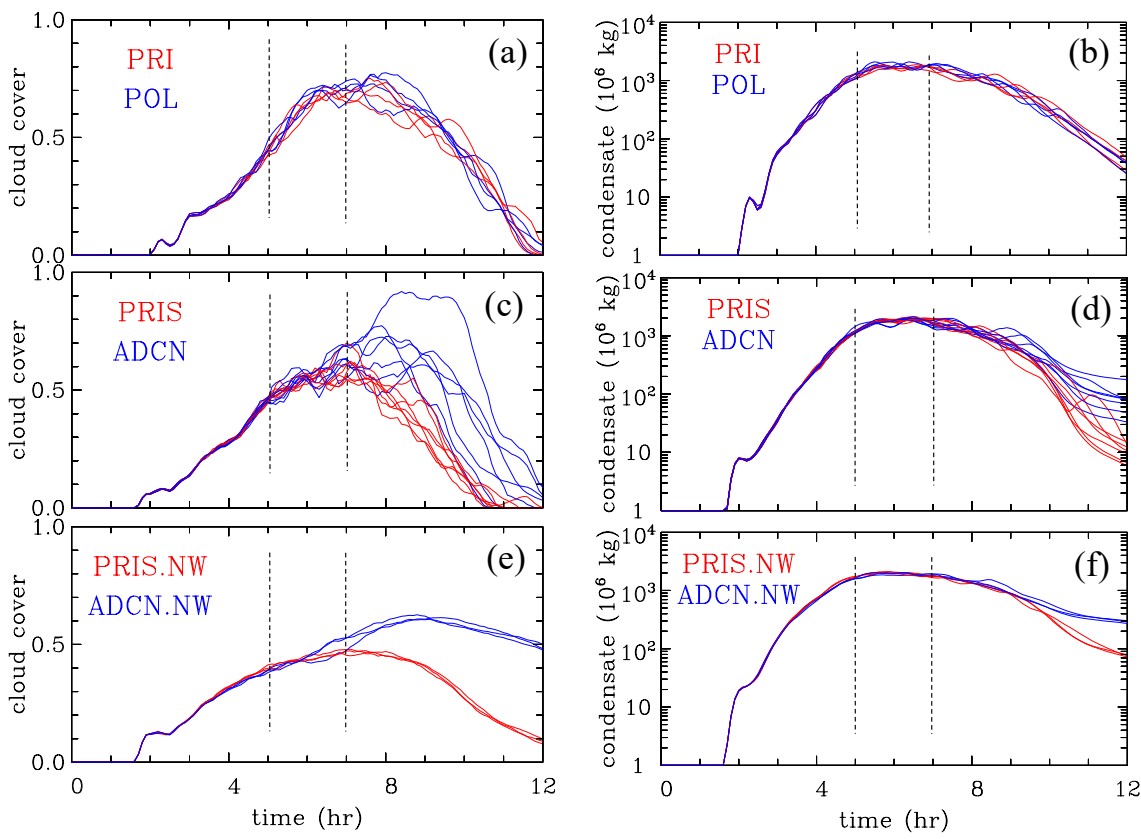

**Figure 2. Evolution of (a, c, e) cloud cover and (b, d, f) total mass of cloud and precipitation inside the computational domain in (a, b) PRI and POL IAB ensembles, (c, d) PRIS and ADCN 2MOM ensembles, and (e, f) PRIS.NW and ADCN.NW 2MOM ensembles. Dashed vertical lines show the 6th and 7th hour period for which analyses are completed.**



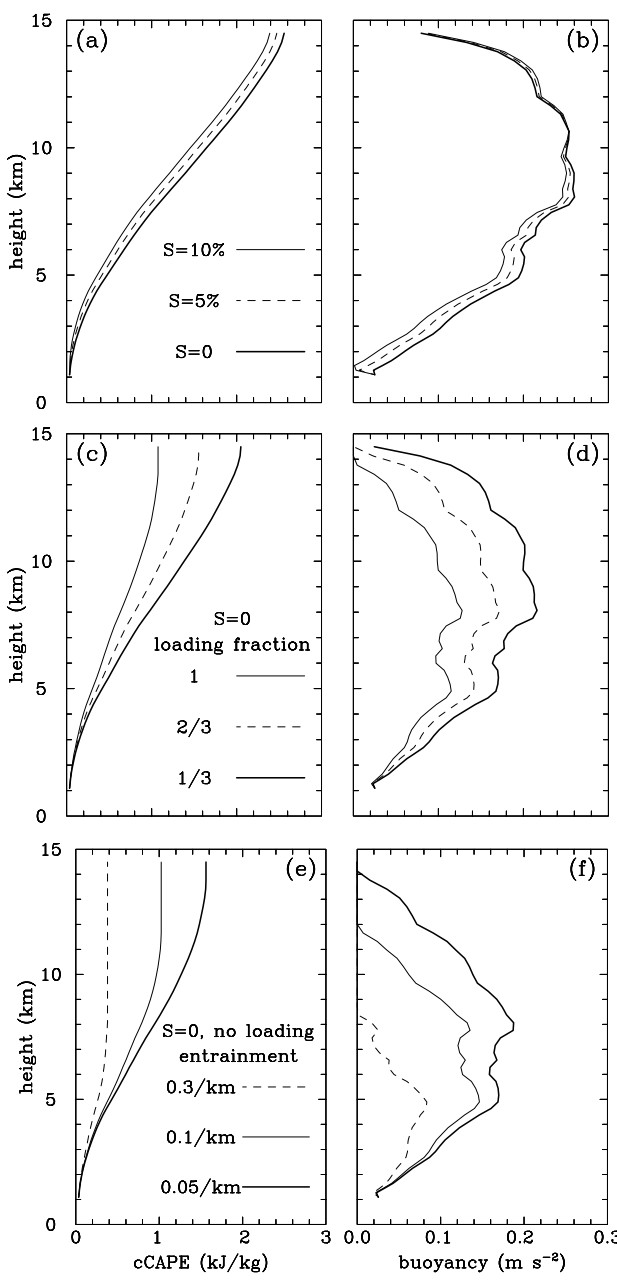


**Figure 3. Profiles of (a, c, e) cumulative CAPE (cCAPE) and (b, d, f) buoyancy from parcel analysis using mean temperature and humidity profiles from the lowest 500 m of 2MOM PRIS simulation averaged over 6th and 7th hour. (a, b) Pseudo-adiabatic parcel calculations with different supersaturations. (c, d) Parcel calculations with different fractions of the condensate included in parcel buoyancy. (e, f) Entraining parcel calculations with different fractional entrainment rates.**




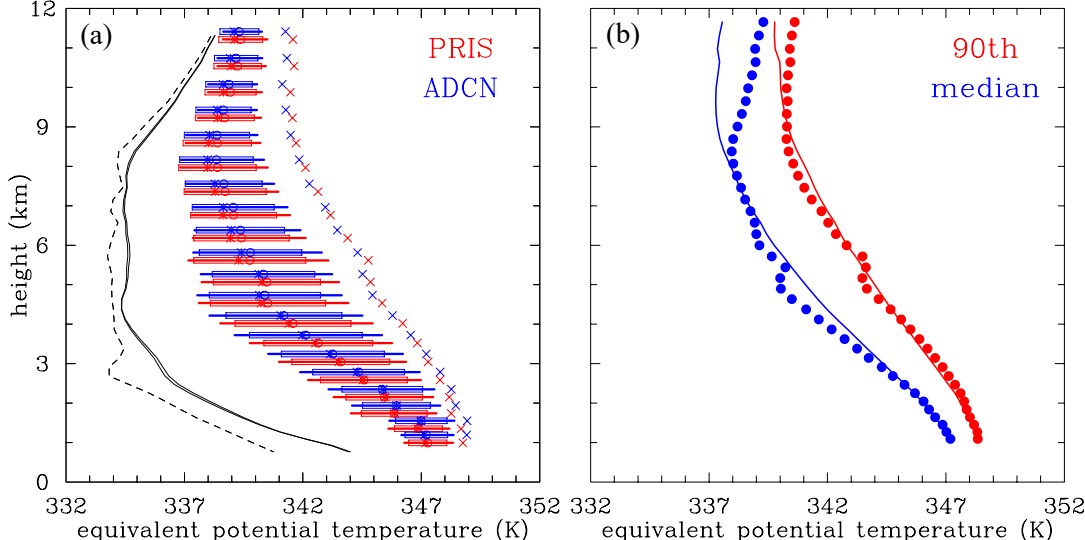


**Figure 4. (a) Comparison of equivalent potential temperature statistics during 6th and 7th hour from GM20 PRIS (red) and ADCN (blue) simulations. Asterisks represent median values, thick lines mark the range between the 10th and 90th percentiles, circles show the mean values, and "x" symbols to the right of the of the boxes represent means of the range between the 90th percentile and the maximum. Boxes represent the range between the mean and plus and minus one standard deviation. Only in-cloud points with**

**vertical velocity larger than 1 m s⁻¹ and total condensate larger than 1 g kg⁻¹ and are included in the statistics. The dashed black line is the equivalent potential temperature profile calculated from the initial sounding. The two solid lines are the equivalent potential temperature profiles calculated from the mean temperature and moisture profiles averaged over 6th and 7th hour from PRIS and ADCN ensemble simulations. (b) Comparison between model and entraining parcel results. Dots are median (blue) and 90th percentile (red) values from the left panel PRIS results. Lines are profiles of the equivalent potential temperature derived from the**

**entraining parcel with the constant entrainment rate that minimizes the difference between the parcel and either the median or the 90th percentile profiles from the model between 1 and 10 km. See text for more details.**



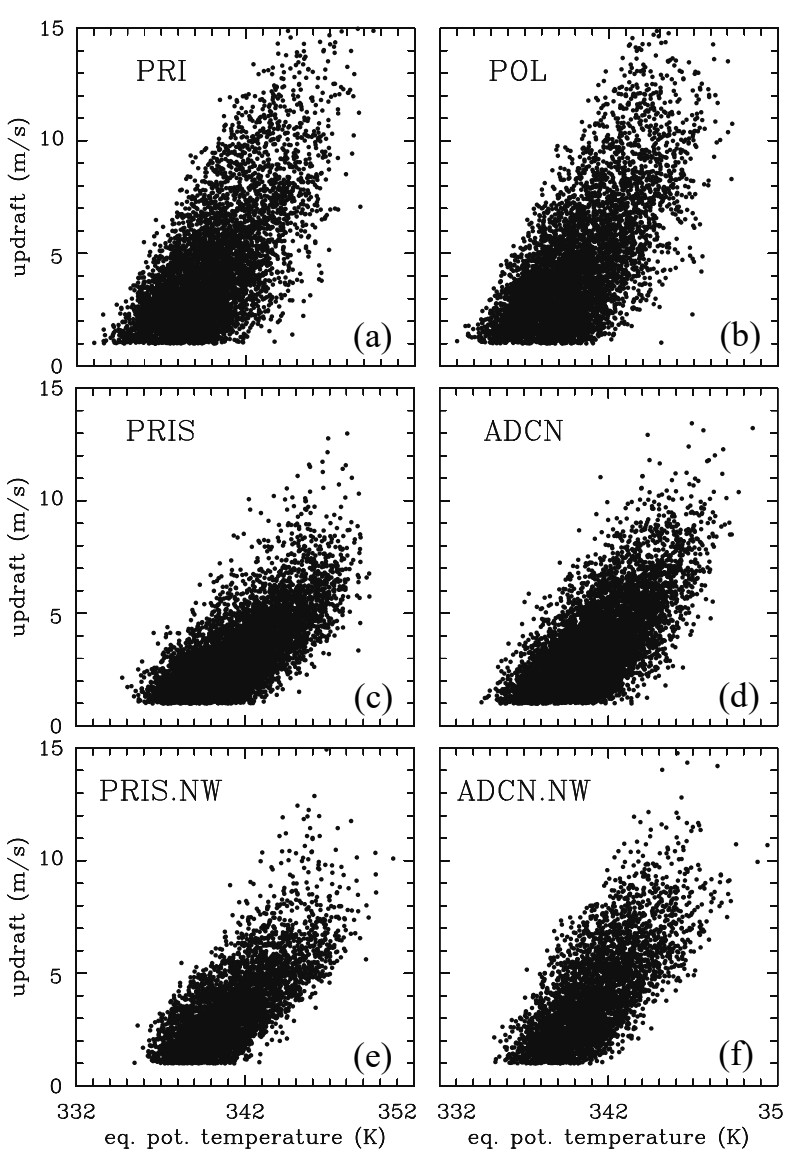

**Figure 5. Updraft versus $\Theta_e$ for 4 IAB simulations (PRI and POL; a and b), 3 2MOM simulations (randomly selected from 7 members for PRIS and ADCN ensembles; c and d), and 3 2MOM simulations from PRIS.NW and ADCN.NW ensembles; e and f). Data at 4km height for hour 6 and 7 in grid volumes with updraft larger than 1 m s$^{-1}$ and total condensate larger than 1 g kg$^{-1}$.**





Figure 6. As Fig 5, but at 9 km for 2MOM (a) PRIS and (b) ADCN.





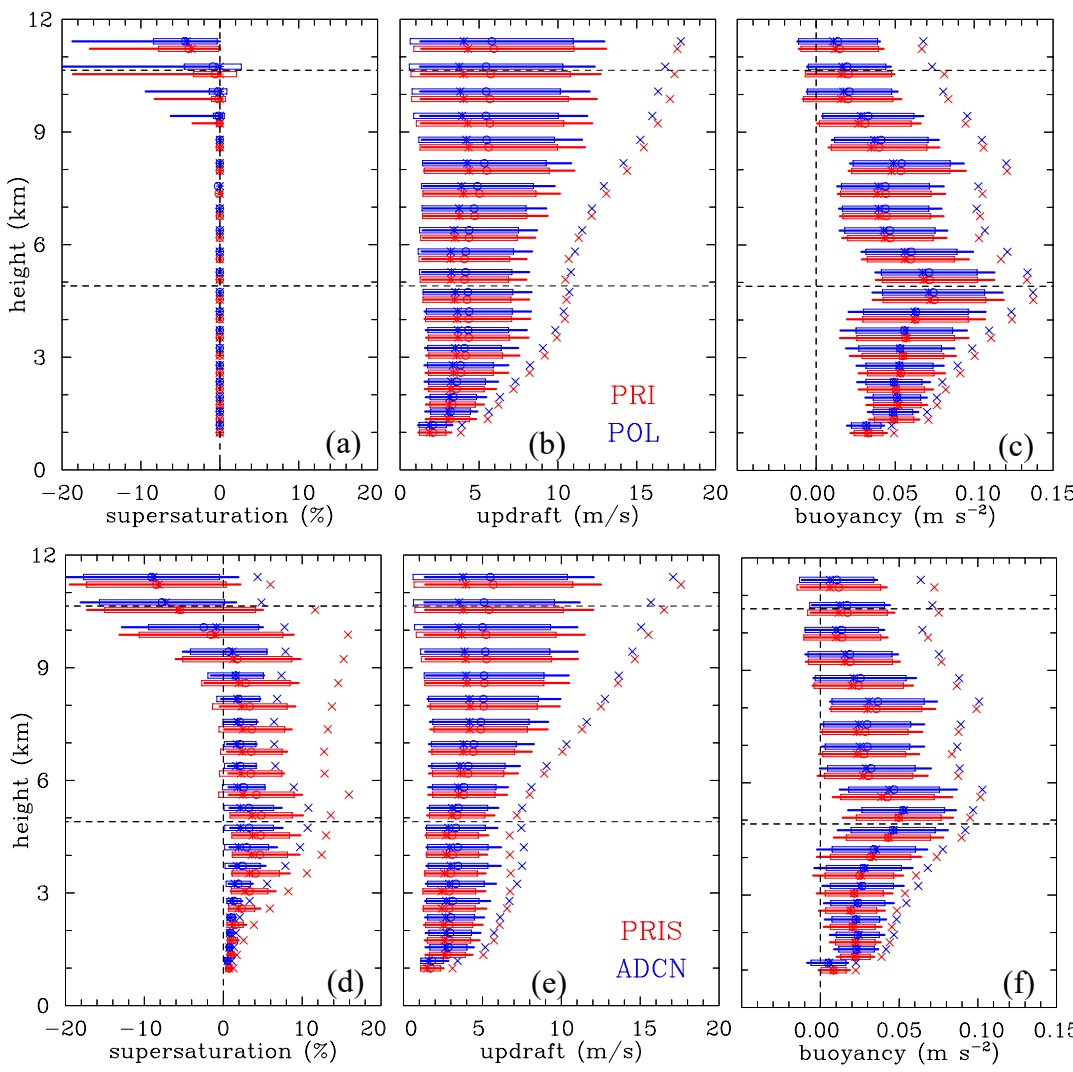

**Fig. 7. Statistics of (a,d) supersaturation, (b,e) vertical velocity, and (c,f) buoyancy in grid volumes with updraft larger than 1 m s⁻¹**
**and total condensate larger than 1 g kg⁻¹. (a,b,c): data for all ensemble members for the 6th and 7th hour of the simulations from PRI**
**(red color) and POL (blue color) IAB ensembles. (d,e,f): data for all ensemble members for PRIS (red color) and ADCN (blue color)**
**for 2MOM ensembles. The star/circle symbols are for median/mean values, horizontal lines show the 10th to 90th percentile range,**
**and the boxes show the range of the mean plus/minus one standard deviation. The x symbols to the right of color lines and boxes are**
**means of the data from the range between the 90th percentile and the maximum. The data are only shown every second model level**
**with color lines and symbols shifted above (for ADCN) and below (for PRIS) that level. Dashed horizontal lines show approximate**
**height of the 0 and -40 degC level. Dashed vertical lines in (a) and (c) show zero values.**





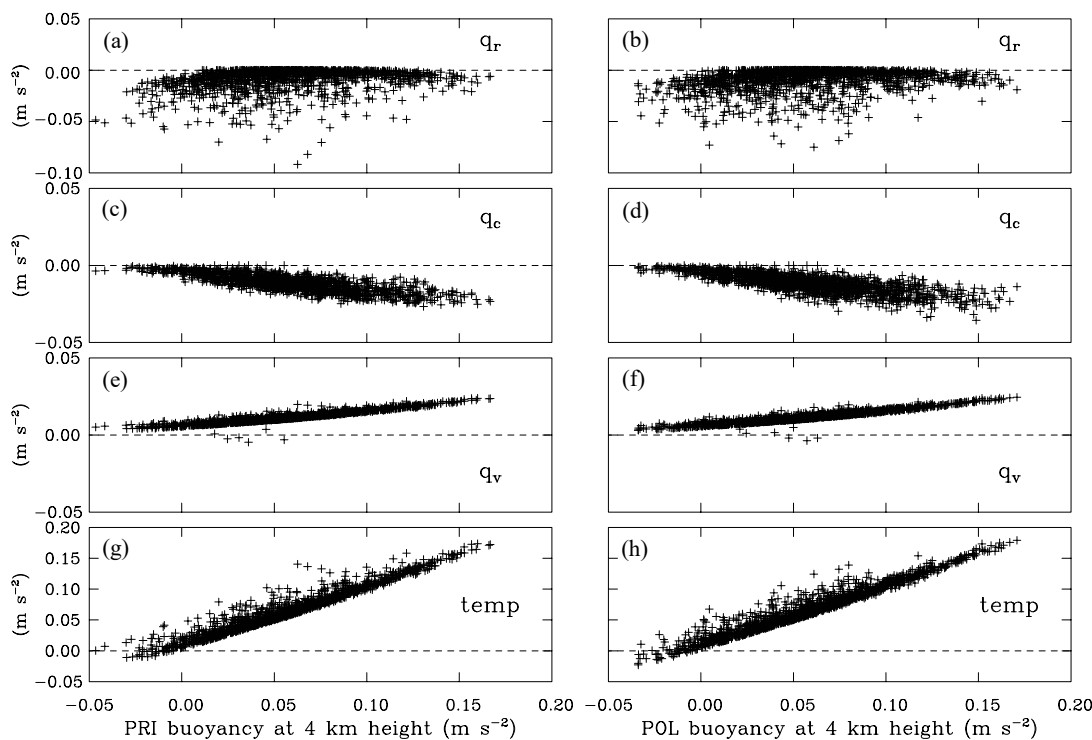


**Figure 8. Scatterplots of buoyancy components at 4 km as a function of the total buoyancy in (a, c, e, g) PRI and (b, d, f, h) POL of all 4 ensemble members of IAB simulations at hour 6 and 7. The horizontal axes in both columns represent the total buoyancy. Panels (g) and (h) show temperature buoyancy component. Panels (e) and (f) show the water vapor contribution. Panels (a), (b), (c) and (d) are the loading buoyancy components, separated into (c, d) cloud water and (a, b) rain components. Point with updraft**
**larger than 1 m s$^{-1}$ and total condensate larger than 1 g kg$^{-1}$ are included. Only about 8% of data points is shown.**







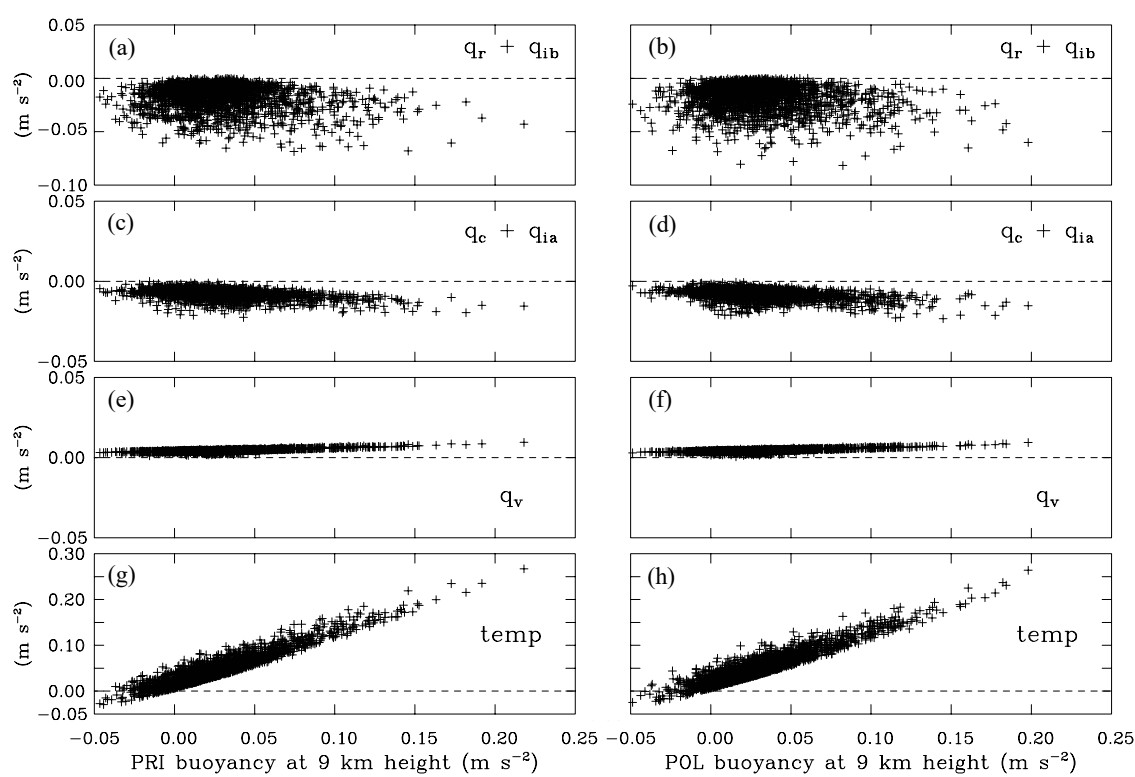


**Figure 9. As Fig. 8, but at 9 km. Panels (a,b) include rain combined with ice B mixing ratios; panels (c,d) include cloud water combined with ice A mixing ratio.**







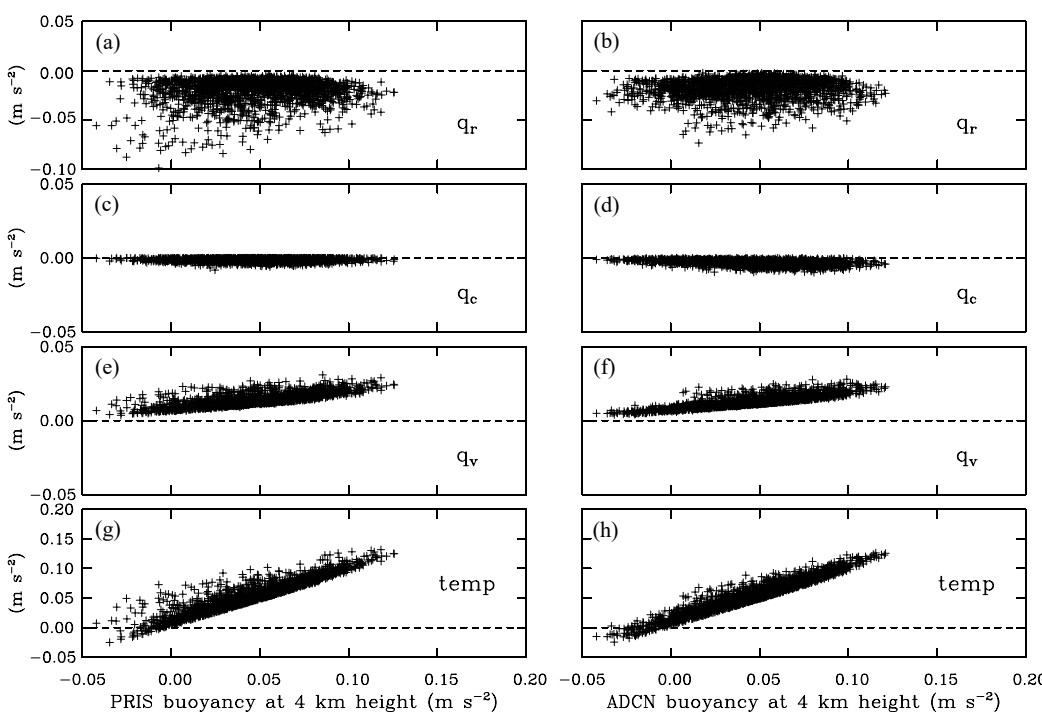

**Figure 10. As Fig. 8 but for all 7 simulations (left panels) PRIS and (right panels) ADCN from 2MOM ensemble. Only 5% of data points are shown.**






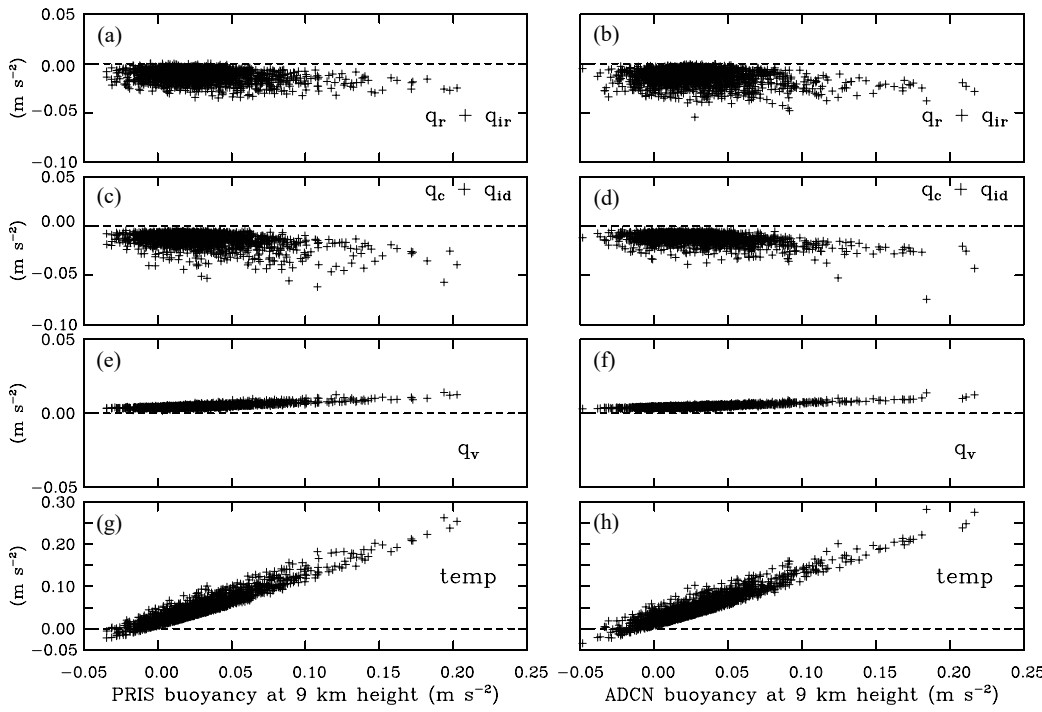

**Fig. 11. As Fig. 10, but at 9 km. Panels (a,b) include rain mixing ratio combined with ice mixing ratio grown by diffusion of water**
**vapor; panels (c,d) include cloud water combined with ice mixing ratio grown by riming.**






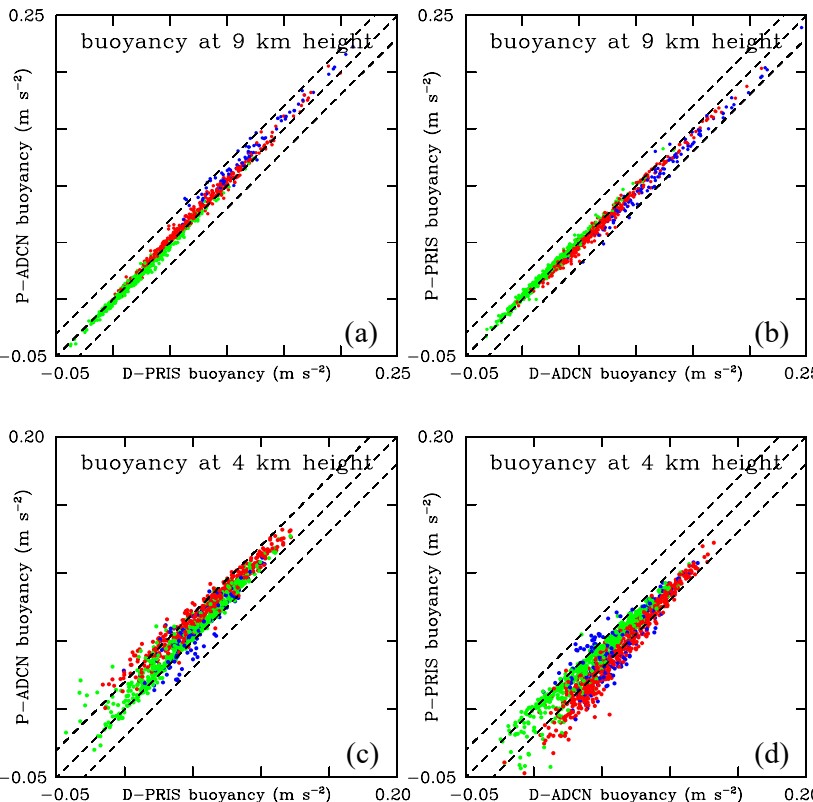

**Figure 12. Driver versus piggybacker buoyancy at (a, b) 9 km and (c, d) 4 km height and hours 6 and 7 for all members of PRIS and**
**ADCN ensembles. Driver buoyancy is shown on the horizontal axes (D-PRIS in a and c; and D-ADCN in b and d). Piggybacker buoyancy is on the vertical axes. Symbol colors represent vertical velocity at the location from which driver and piggybacker buoyancies are taken. At 4 km, green, red, and blue represent updrafts between 1 and 5 m/s, 5 to 10 m/s, and above 10 m/s, respectively, with only 2% of all points shown for green, 10% for red, and all for blue. At 9 km, green, red, and blue colors represent updrafts between 1 and 10 m/s, 10 to 20 m/s, and above 20 m/s, with 4% of all points shown for green, 20% for red, and all for blue.**
**Middle dashed lines show equal buoyancies, and the lines above and below show buoyancies offset by 0.02 m s⁻². Only points with vertical velocity larger than 1 m/s and total condensate mixing ratio larger than 1 g/kg are included in the plot.**




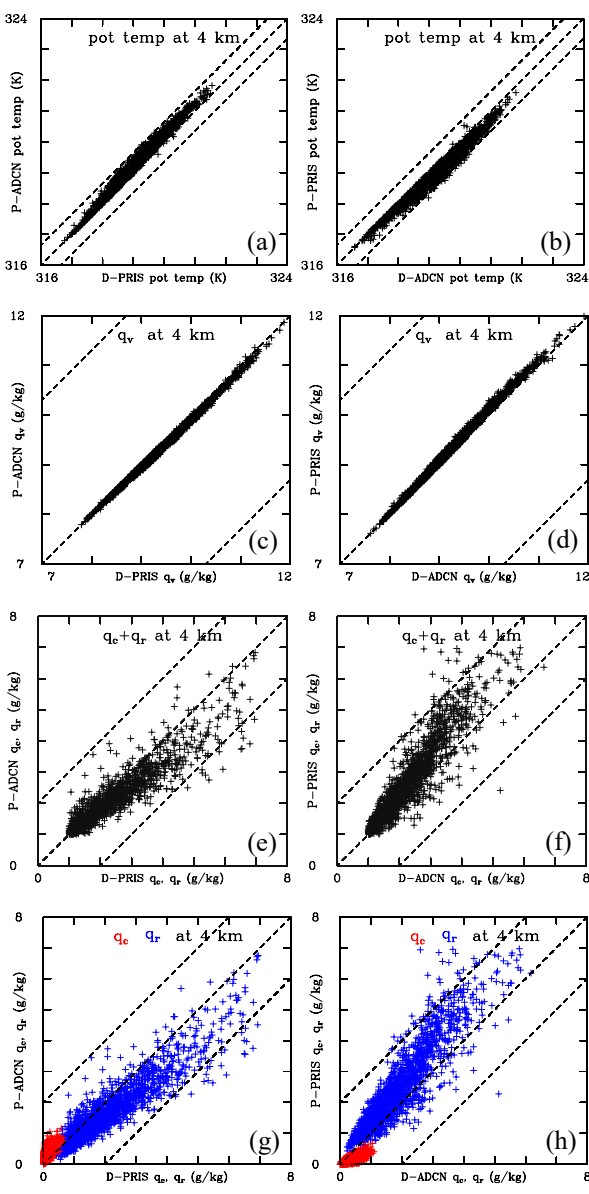

**Figure 13. Driver versus piggybacker for components of the buoyancy at 4 km shown in lower panels of Fig. 12. (a,b) – temperature; (c,d) – water vapor; (e,f) – total loading; (g,h) – loading split into cloud water (red) and rain (blue). Dashed lines below and above the 1:1 dashed line in all panels correspond to the buoyancy impact as shown by dashed line in Fig. 12. Only 5% of data points with the vertical velocity larger than 1 m s⁻¹ and the total condensate larger than 1 g kg⁻¹ are used.**



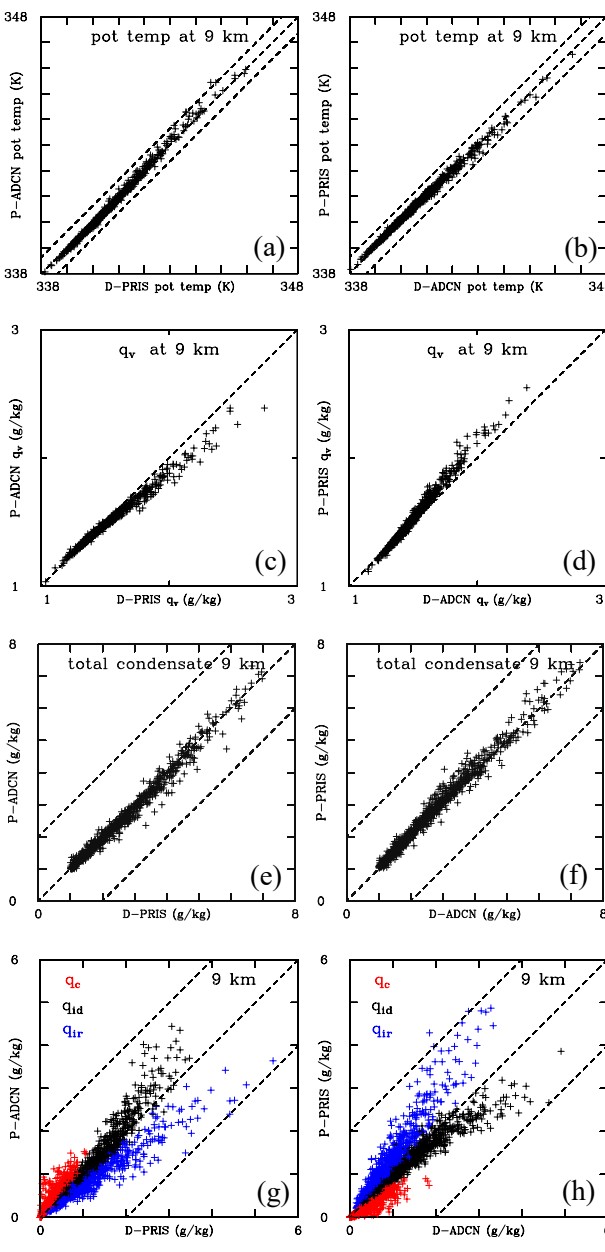

**Figure 14. As in Fig. 13, but for upper panels of Fig. 13, that is, at 9 km height. Dashed lines for qv (middle panels) are outside the range of qv values shown in the figure. The total loading component in (e,f) is split in (g.h) between cloud water (red), ice mixing ratio grown by diffusion of water vapor (black), and ice mixing ratio grown by riming (blue).**