# Peer review of "Supersaturation, buoyancy, and deep convection dynamics"

_Atmospheric Chemistry and Physics, 2021_

## Author Comment (AC1)

Responses to the Reviewer 2 comments.

The authors appreciate thoughtful comments from the Reviewer 2. Below we reply to those comments and list revisions that will be included in the revised manuscript. The original comment text is in black, our responses are in red.

This manuscript combined theoretical and modeling analysis to investigate the factors that influence the buoyancy of convective updrafts including the in-cloud supersaturation, condensate and precipitation loading and entrainment. The authors leverage an ensemble of idealized simulations of varying complexity they have performed during previously published work but provide additional detailed analysis to quantify the relative contributions to buoyancy generation and loss. While the work is motivated by the continuing debate centered around aerosol invigoration of deep convection, to which the authors have made significant contributions, the analysis and results have broader implications to our general understanding of deep convective core dynamics. Compliments to the authors on a very clearly written manuscript. In my opinion, the manuscript is nearly ready for publication in ACP, but would benefit from some minor changes for clarity in a few places.

Opportunities for clarification:

Last paragraph of Introduction – This paragraph is mostly a listing of the different model ensemble members that are incorporated in the analysis. These differences are important for the reader to understand, but I found it quite difficult to parse what they all were. I think a table, or even a bulleted or numbered list might help to present these more clearly.

We think the reviewer means the last paragraph of section 2.1. We added a table listing all ensembles and describing aspects discussed in this section.

Line 225 – While Figure 1 shows equivalent potential temperature, the test refers to moist static energy. While these are similar measures, consistency in terms will help the reader. The similarity of equivalent potential temperature and moist static energy is explained in line 251-252.

We modified the Fig. 1 caption to clearly explain the link between the equivalent potential temperature and the moist static energy.

Figure 5 and 6 would benefit greatly from including a grid such that the reader can more easily identify the features of the comparison that are discussed in the text.

We modified the Figs. 5 and 6 per this suggestion.

Line 480 – Make it clear that the right and left panels of Fig. 8 and 9, that are being referred to, are the pristine and polluted cases, respectively.

We modified the text to stress that.

Minor comments:

Line 130 – "….have to come from different reasons…" I think I know what is intended here, but this phrase does not quite make sense. Would something like, "…require different explanations.."?

Line 197 – "rimes" should be "rimed"

Line 225 – "the cloud cover in left panels" should be "the cloud cover in the left panels"

Line 235 – It would be helpful to add "ice crystal" before sizes. At least I think that is the sizes that are being referred to.

Line 257 – Add "the" between "obtain" and "buoyancy"

Line 619 – Add "a" before "more"

Line 647 – "set" should be "sets"

Line 655 – "it" should be "its"

All the above suggestions and typos were corrected.

---

## Author Comment (AC3)

Responses to the Reviewer 1 comments.

The authors appreciate thoughtful comments from the Reviewer 1. Below we reply to those comments and outline revisions that will be included in the revised manuscript. The original comment text is in black, our responses are in red. However, the review as published on the ACPD webpage is poorly organized, so we took the entire text without any changes and put it below with all comments sequentially numbered.

This paper deals with the impact of supersaturation, condensate and precipitation loading, and entrainment on parcel buoyancy for moist deep convection. The authors analyze those factors using both a 1D parcel model and 3D cloud-resolving simulations to conclude that supersaturation can suppress deep convection, especially in the lower troposphere. However, both water/precip loading and entrainment have a more significant cumulative impact across the entire troposphere. Additional tests for either pristine or polluted air conditions indicate only minor changes due to microphysics.

I find this study an interesting and important contribution to our improved understanding of deep convective dynamics. Small-scale models and convection parameterizations are often agnostic to supersaturation-buoyancy feedbacks, which may potentially deteriorate their simulation of deep convection for certain scenarios. The paper is quite well written, and the conclusions are supported by a convincing set of analyses. However, some parts of it would benefit from a better structure and additional explanations, as suggested below.

Comments:

1. Title: Since deep convection is the focus of the paper, "deep convective dynamics" would better reflect its content.

We agree. The title has been changed to "… deep convection dynamics".

2. Abstract: it has almost 400 words and includes (too) many details. I suggest to shorten it (~250 words) and make a take home message more succinct.

The abstract has been shortened to about 300 words.

3. What is the impact of your anelastic approximation on the simulation of deep convection? Apart from the lack of baroclinicity in your equations, and a simplified form of continuity equation, the buoyancy term is normalized on the arbitrarily-chosen base state temperature and your results may differ from the most accurate fully compressible solution.

We do not want to discuss anelastic versus compressible solutions as this has been addressed in previous studies, for instance, in Kurowski et al., *J. Atmos. Sci.* 2013, 2014, and 2015. The fully-compressible equations (i.e., with gravity term as "… + g + …) are seldom used in the atmospheric dynamics, and the Boussinesq form of compressible equations (… + g rho'/rho + …) does include a hydrostatically-balanced background state (through rho). For instance, please compare equation sets in sections 2a and 2b in Bryan and Fritsch (*Mon. Wea. Rev.* 2002). The Boussinesq form is sometimes referred to as the reduced gravity method.

No changes to the manuscript in response to this comment.

4. Your latent heat of condensation is assumed constant while in reality it somewhat depends on temperature (for this range of temperatures it may vary by several percent) – is this effect important for the calculated buoyancy?

This comment is only partially true. Both IAB and 2MOM schemes include temperature-dependent latent heat of condensation. I think the reviewer was confused by the "=" sign in the second paragraph of the introduction. We changed it to "≈" there. Parcel analysis indeed assumes a constant latent heat of condensation. Our tests show that this has a small (below 10%) impact on actual values of cCAPE and other quantities derived in the analysis. For instance, total CAPE is smaller when variable latent heat is assumed because latent heating is reduced in the lower troposphere where the latent heating is the smallest. We decided not to bring this subject in the revised text as it is only marginally relevant to the main thrust of the paper.

No changes to the manuscript in response to this comment.

5. How strongly do your results depend on model resolution? Your dx=400m is quite coarse and one may expect higher vertical velocities (and supersaturations) for finer resolution simulations.

Not necessarily. Higher resolution implies more resolved entrainment and thus may have the opposite effect. Grabowski and Prein (*J. Climate*, 2019) compared simulations as in the paper under review and shorter higher resolution (truly LES-type) simulations applying a modified LBA case. Comparison of Figs. 7 and 14 there shows that the cloud fraction profiles evolve similarly in the lower and higher resolution simulations, at least up to hour 4 of the simulations. The impact of entrainment is mentioned in the paragraph starting in line 86 in the introduction. We feel this sufficient.

No changes to the manuscript in response to this comment.

6. How does supersaturation affect mass flux? You show in Fig. 7 different vertical velocities with/without S, but what happens to the area?

Cloud fraction profiles are only weakly affected before significant anvils develop as shown in our previous papers. For instance, see Fig. 4 in G15, Fig. 1 in Grabowski and Morrison (*J. Atmos. Sci.* 2016), and Fig. 1 in GM20. Arguably, Fig. 2 in the current paper documents that as well. Since the paper focuses on the convective dynamics, buoyancy and updraft strength in particular, we do not want to discuss this aspect.

No changes to the manuscript in response to this comment.

7. I don't see a clear justification for using IAB in this study and thus removing it may help in keeping the main message clearer. Instead of comparing G15 and GM20, can you simply use GM20 with and without supersaturation (S=0 for the latter) to directly evaluate its impact?

Keeping IAB and 2MON ensembles allows comparing results with not only saturation adjustment and supersaturation prediction, but also significantly different microphysics parameterizations.

We will add a comment on the in the revised text.

8. Section 3: Can you clarify the purpose of your analysis at the beginning of this section and explain how it relates to your supersaturation considerations? d/dz, psi, and psi_e can all be obtained from the 3D model output, so what exactly do you want to calculate?

The purpose of this analysis is to compare idealized entraining parcel calculations with the dynamic model simulations. The parcel results are included in the Table 2 and compared with the dynamic model results in Fig. 4. We feel this is a useful element of the manuscript.

We will add a comment on the in the revised text.

9. Provide the definition of equivalent potential temperature here.

DO IT…

10. Since rising plumes represent the right tails of moisture and temperature distributions near the surface, using 15-20% of that distribution instead of mean values in the lowest 500m as the initial conditions may be more relevant. That would be similar to what convection parameterizations do, e.g.:
https://journals.ametsoc.org/view/journals/atsc/76/8/jas-d-18-0239.1.xml Have you looked at the impact of initial conditions on your parcel model results?

It is a traditional approach to consider the lowest-500-meter-averaged properties to derive convective indices. In response to this comment, we ran parcel simulations with initial conditions taken as surface values. These values are about 0.1 K warmer and about 0.5 g/kg more humid that 500-meter averages. The outcome – as one might expect – is significant for specific values shown in the Table 1 in the original submission (Table 2 in the revised text), with some parameters (e.g., CAPE) increasing by up to 20%. However, the relative differences between various parcel simulations remain similar. We mention this in the revised text.

11. There are LES studies (see below; also for LBA) showing that rising plumes/thermals can reach a quasi-steady velocity due to the balance between buoyancy and drag, which may be an explanation of the differences between the theoretical sqrt(2CAPE) and your simulation results.
https://journals.ametsoc.org/view/journals/atsc/73/10/jas-d-15-0385.1.xml

This is a valid point. The paper the reviewer refers to argues that thermals are more appropriate than plumes as building blocks of deep convection. We agree with such an argument. However, it is not clear to us what the reviewer suggests us to do. We assume that this comment is in regard to the statement on lines 291-294 in the original manuscript that vertical velocities in the simulation are 2-3 times smaller. Besides drag as the reviewer mentions (or, more generally, an adverse vertical perturbation pressure gradient force), the impact of entrainment and condensate loading likely plays a role in explaining the difference between the theoretical and simulated vertic al velocities. All of these factors are mentioned in that sentence. Since drag is a component of the perturbation pressure forcing already mentioned in this sentence, we have not modified the text.

12. In Fig. 12, the spread of your results is larger at 4km than at 9km. Is it because the updrafts are more separated from the environment at higher velocities?

We do not think so. We think this is because of the latent heating differences between pristine and polluted conditions that are more significant at 4 km than at 9 km. Please note that the equivalent potential temperature at 9 km is closer to the environment with less scatter than at 3 km. This can arguably explain differences in the statistics shown in Fig. 12.

We will add a comment on the in the revised text.

13. Is piggybacking only useful to look at tiny effects due to microphysics or is it a more universal method?

We feel piggybacking can be used to study impact of any element of the model physics. Grabowski and Prein (*J. Climate*, 2019) compared the impact of different temperature and moisture profiles on convective development in the context of the climate change. Kurowski et al. (*Geophys. Res. Lett.* 2019) applied piggybacking to study the impact of environmental heterogeneities (e.g., remnants of previous clouds) in shallow convection simulations. One can think of various other processes that can be studied using piggybacking, such as radiative transfer, surface heat fluxes, etc.

14. In Conclusions, "unorganized deep convection" – this statement is questionable. When cold pools are not present, buoyancy-driven plumes can only reach up to ~9km for this case. Your updrafts reach to the top of troposphere (14-15km) as for organized deep convection although autocorrelation scale may be limited by the size of your domain. You could actually cite this paper https://journals.ametsoc.org/view/journals/atsc/75/12/jas-d-18-0031.1.xml around the discussion of the LBA setup. Even for the 50km domain, your convection reaches the tropopause, as for larger-domain simulations.

We feel this is a misunderstanding. What we mean is that the convention is scattered, that is, there are no squall lines, bow echos, or other organized convection systems. We will change "unorganized" into "scattered" in the revised text.

15. Your conclusions mostly focus on invigoration (mentioned 7x), whereas they should also describe briefly your analysis results, that is the impact of supersaturation, water loading, and entrainment on the buoyancy.

We will revise the conclusion section following this suggestion.

16. Please mention about some 1D convection parameterizations based on an entraining plume approach. Typically, they additionally employ a steady-state velocity equation affected by buoyancy, e.g.:
https://agupubs.onlinelibrary.wiley.com/doi/full/10.1002/2015MS000502
https://journals.ametsoc.org/view/journals/atsc/76/8/jas-d-18-0239.1.xml

We do not think including convection parameterization aspect is needed in this already quite a long manuscript.

No change to the text is response to this comment.

---

## Author Response (AR2)

**Responses to the Reviewer 1 additional comments.**

*Reviewer's comments in black, our original responses in red, responses and revisions as a result of the additional comments in green.*

The authors' response is minimalist. I understand their arguments but my intention was to ask important questions that the reader may ask as well to get a better understanding of the presented work. Ultimately, you want to convince the reader that the methodology you applied represents the reality with high fidelity.

I suggest that the authors add 1-2 sentences to the manuscript with at least minimal explanations regarding my points 3, 4, 5, 6, 13, 14.

Please cite this paper:
https://agupubs.onlinelibrary.wiley.com/doi/abs/10.1029/2021GL093804
to strengthen your conclusions.

We agree that our responses and revisions were minimalist, but we felt this was the best approach for the paper. Because the Reviewer seems unsatisfied with our approach, we added text as requested, mostly through several footnotes. Below, our original responses to the points 3, 4, 5, 6, 13, 14 are in red, and responses and revisions are explained in green. References to the recent paper suggested by the reviewer is added to the conclusion section.

3. What is the impact of your anelastic approximation on the simulation of deep convection? Apart from the lack of baroclinicity in your equations, and a simplified form of continuity equation, the buoyancy term is normalized on the arbitrarily-chosen base state temperature and your results may differ from the most accurate fully compressible solution.

We do not want to discuss anelastic versus compressible solutions as this has been addressed in previous studies, for instance, in Kurowski et al., *J. Atmos. Sci*. 2013, 2014, and 2015. The fully-compressible equations (i.e., with gravity term as "… + g + …) are seldom used in the atmospheric dynamics, and the Boussinesq form of compressible equations (… + g rho'/rho + …) does include a hydrostatically-balanced background state (through rho). For instance, please compare equation sets in sections 2a and 2b in Bryan and Fritsch (*Mon. Wea. Rev*. 2002). The Boussinesq form is sometimes referred to as the reduced gravity method.

No changes to the manuscript in response to this comment.

We added a footnote in the introduction, just above the Eq. 1. The footnote reads:

"For a discussion of the anelastic versus compressible equations and simulation results obtained from the two in the context of small-scale and planetary-scale dynamics, the reader is referred to Kurowski et al. (2013, 2014, 2015) and references therein."

We added Kurowski et al. information to the list of references.

4. Your latent heat of condensation is assumed constant while in reality it somewhat depends on temperature (for this range of temperatures it may vary by several percent) – is this effect important for the calculated buoyancy?

This comment is only partially true. Both IAB and 2MOM schemes include temperature-dependent latent heat of condensation. I think the reviewer was confused by the "=" sign in the second paragraph of the introduction. We changed it to "≈" there. Parcel analysis indeed assumes a constant latent heat of condensation. Our tests show that this has a small (below 10%) impact on actual values of cCAPE and other quantities derived in the analysis. For instance, total CAPE is smaller when variable latent heat is assumed because latent heating is reduced in the lower troposphere where the latent heating is the smallest. We decided not to bring this subject in the revised text as it is only marginally relevant to the main thrust of the paper.

No changes to the manuscript in response to this comment.

We added a footnote to the paragraph below Eq. 2. The footnote reads:

"The code for parcel calculations applies a constant latent heat of condensation in contrast to the microphysical schemes applied in the dynamic model. This has a small (below 10%) impact on actual values of cCAPE and other quantities derived in the analysis. For instance, total CAPE is smaller when variable latent heat of condensation is assumed because latent heating is reduced in the lower troposphere where the latent heat of condensation is the smallest."

5. How strongly do your results depend on model resolution? Your dx=400m is quite coarse and one may expect higher vertical velocities (and supersaturations) for finer resolution simulations.

Not necessarily. Higher resolution implies more resolved entrainment and thus may have the opposite effect. Grabowski and Prein (*J. Climate*, 2019) compared simulations as in the paper under review and shorter higher resolution (truly LES-type) simulations applying a modified LBA case. Comparison of Figs. 7 and 14 there shows that the cloud fraction profiles evolve similarly in the lower and higher resolution simulations, at least up to hour 4 of the simulations. The impact of entrainment is mentioned in the paragraph starting in line 86 in the introduction. We feel this sufficient.

No changes to the manuscript in response to this comment.

We added the following text to the second paragraph in section 2.1:

"Overall, the horizontal resolution is relatively low making the simulations only marginally LES, especially early in the simulations when the boundary layer is relatively shallow. However, as mentioned in G15 (section 2a therein) applying such a grid provides results broadly consistent with the high-resolution benchmark simulations reported in Grabowski et al. (2006). Results reported here seem also consistent with truly-LES simulations reported in Kurowski et al. (2018) and in Grabowski and Prein (2019)."

6. How does supersaturation affect mass flux? You show in Fig. 7 different vertical velocities with/without S, but what happens to the area?

Cloud fraction profiles are only weakly affected before significant anvils develop as shown in our previous papers. For instance, see Fig. 4 in G15, Fig. 1 in Grabowski and Morrison (*J. Atmos. Sci.* 2016), and Fig. 1 in GM20. Arguably, Fig. 2 in the current paper documents that as well. Since the paper focuses on the convective dynamics, buoyancy and updraft strength in particular, we do not want to discuss this aspect.

No changes to the manuscript in response to this comment.

The following short paragraph was added at the end of section 4.2:

"Although not directly relevant to the main thrust of this paper, it is worthwhile to mention that the convective mass flux is also insignificantly affected by small differences in the convective dynamics documented in Fig. 7. This is because cloud fraction profiles are only weakly affected by microphysical processes (at least before significant anvils develop in 2MOM simulations) as shown in G15 (Fig. 4 therein), Grabowski and Morrison (2016; Fig. 1 therein), and GM20 (Fig. 1 therein). Arguably, Fig. 2 herein documents that as well.

13. Is piggybacking only useful to look at tiny effects due to microphysics or is it a more universal method?

We feel piggybacking can be used to study impact of any element of the model physics. Grabowski and Prein (*J. Climate*, 2019) compared the impact of different temperature and moisture profiles on convective development in the context of the climate change. Kurowski et al. (*Geophys. Res. Lett.* 2019) applied piggybacking to study the impact of environmental heterogeneities (e.g., remnants of previous clouds) in shallow convection simulations. One can think of various other processes that can be studied using piggybacking, such as radiative transfer, surface heat fluxes, etc.

The following text was added as a footnote in section 4.3:

"Piggybacking can be used to study impact of any element of the model physics. Grabowski and Prein (2019) compared the impact of different temperature and moisture profiles on convective development in the context of climate change. Kurowski et al. (2019) applied piggybacking to study the impact of environmental heterogeneities (e.g., remnants of previous clouds) in shallow convection simulations. Impacts of various other processes can be studied using piggybacking, such as radiative transfer, surface heat fluxes, etc. See Grabowski (2019)."

14. In Conclusions, "unorganized deep convection" – this statement is questionable. When cold pools are not present, buoyancy-driven plumes can only reach up to ~9km for this case. Your updrafts reach to the top of troposphere (14-15km) as for organized deep convection although autocorrelation scale may be limited by the size of your domain. You could actually cite this paper https://journals.ametsoc.org/view/journals/atsc/75/12/jas-d-18-0031.1.xml around the discussion of the LBA setup. Even for the 50km domain, your convection reaches the tropopause, as for larger-domain simulations.

We feel this is a misunderstanding. What we mean is that the convention is scattered, that is, there are no squall lines, bow echoes, or other organized convection systems. We will change "unorganized" into "scattered" in the revised text.

We replaced "unorganized" with "scattered" in the revised text. We added reference to the Kurowski et al. (2018) paper.